# Prostaglandin E$_2$ receptor Ptger4b regulates female-specific peptidergic neurons and female sexual receptivity in medaka

Thomas Fleming [1], Yukiko Kikuchi[1], Mikoto Nakajo [1,2], Masaya Tachizawa[1], Tomoaki Inazumi[3], Soken Tsuchiya[3], Yukihiko Sugimoto [3], Daisuke Saito[4], Mikita Suyama [4], Yasuyuki Ohkawa [5], Takashi Baba [6], Ken-ichirou Morohashi [6] & Kataaki Okubo [1✉]

In vertebrates, female receptivity to male courtship is highly dependent on ovarian secretion of estrogens and prostaglandins. We recently identified female-specific neurons in the medaka (*Oryzias latipes*) preoptic area that express Npba, a neuropeptide mediating female sexual receptivity, in response to ovarian estrogens. Here we show by transcriptomic analysis that these neurons express a multitude of neuropeptides, in addition to Npba, in an ovarian-dependent manner, and we thus termed them female-specific, sex steroid-responsive peptidergic (FeSP) neurons. Our results further revealed that FeSP neurons express a prostaglandin E$_2$ receptor gene, *ptger4b*, in an ovarian estrogen-dependent manner. Behavioral and physiological examination of *ptger4b*-deficient female medaka found that they exhibit increased sexual receptivity while retaining normal ovarian function and that their FeSP neurons have reduced firing activity and impaired neuropeptide release. Collectively, this work provides evidence that prostaglandin E$_2$/Ptger4b signaling mediates the estrogenic regulation of FeSP neuron activity and female sexual receptivity.

[1] Department of Aquatic Bioscience, Graduate School of Agricultural and Life Sciences, The University of Tokyo, Tokyo 113-8657, Japan. [2] Department of Physiology, Osaka Medical and Pharmaceutical University, Takatsuki 569-8686, Japan. [3] Department of Pharmaceutical Biochemistry, Graduate School of Pharmaceutical Sciences, Kumamoto University, Kumamoto 862-0973, Japan. [4] Division of Bioinformatics, Medical Institute of Bioregulation, Kyushu University, Fukuoka 812-8582, Japan. [5] Division of Transcriptomics, Medical Institute of Bioregulation, Kyushu University, Fukuoka 812-8582, Japan. [6] Department of Molecular Biology, Graduate School of Medical Sciences, Kyushu University, Fukuoka 812-8582, Japan. ✉email: okubo@marine.fs.a.u-tokyo.ac.jp

In many species, females are sexually receptive to males only during a period surrounding ovulation to ensure reproductive success. In vertebrates, this is accomplished by the proper control of sexual receptivity by ovarian secretion of estrogens, progestins, and prostaglandins, in addition to appropriate sensory cues from males[1,2]. Studies in rodents have revealed that these ovarian hormones act on a hypothalamic-limbic circuit, comprising the arcuate, medial preoptic, and ventromedial nuclei of the hypothalamus and the medial amygdala, to facilitate sexual receptivity to courting males[3,4]. Because these nuclei are phylogenetically ancient and somewhat molecularly conserved, it is assumed that this behaviorally relevant circuit is shared among vertebrates[5,6]. However, little information is available on the neural basis of female sexual receptivity in non-rodent species, and moreover, accumulating evidence highlights large variations in the hormonal regulation of vertebrate mating behavior across taxa. For example, estrogens have stimulatory effects on both male- and female-typical mating behaviors in rodents but prevent the execution of male-typical mating behavior in quail and a teleost species, medaka (Oryzias latipes)[7,8]. In several other teleost species, including goldfish (Carassius auratus) and African cichlid (Astatotilapia burtoni), gonadal estrogens are not essential for female sexual receptivity, and instead, prostaglandin $F_{2\alpha}$ ($PGF_{2\alpha}$) facilitates female receptivity[1,9]. Although it is not known whether other prostaglandin species, such as prostaglandin $E_2$ ($PGE_2$), also centrally regulate female receptivity in teleosts, $PGE_2$ has been shown to have facilitative effects on female receptivity in rats and hamsters but have inhibitory effects in guinea pigs and anole lizards (Anolis carolinensis)[10]. These lines of evidence suggest that there may be a degree of underlying variation—at either the structural or chemical level—in behaviorally relevant circuits and the action of hormonal mediators therein across species.

In medaka, we have recently identified a group of estrogen-dependent neurons relevant to female sexual receptivity, which have not been found in rodents or other species. These neurons are present only in females in the preoptic nucleus PMm/PMg (magnocellular/gigantocellular portion of the magnocellular preoptic nucleus), which is considered homologous to the paraventricular nucleus (PVN) in mammals[11]. Ovarian estrogens regulate the number and size of these neurons, as well as their expression of a gene encoding neuropeptide B (npba)[11,12]. Female medaka deficient for npba and its receptor gene, npbwr2, exhibit abnormal sexual receptivity[13]. Moreover, females deficient for esr2b, a subtype of estrogen receptor expressed in these neurons, show a marked decrease in the number and size of these neurons and a complete loss of sexual receptivity[8]. These findings indicate that npba-expressing neurons in the PMm/PMg represent a major site of action for estrogens on the neuronal circuitry governing female receptivity in medaka.

In the present study, we defined the ovarian secretion-dependent transcriptome of female-specific npba-expressing neurons in the PMm/PMg, which revealed that they express multiple neuropeptides, in addition to Npba, depending on ovarian secretion. We further found that these neurons express a subtype of $PGE_2$ receptor, Ptger4b, in an ovarian estrogen-dependent manner and that $PGE_2$/Ptger4b signaling has an inhibitory effect on female sexual receptivity. This effect was likely due to modulation of the electrophysiological properties of these neurons by Ptger4b, which ultimately govern neuropeptide release.

## Results

### Female-specific preoptic neurons express multiple neuropeptides in an ovarian-dependent manner.
In order to molecularly define female-specific npba-expressing neurons in the PMm/PMg, we searched for genes that are expressed in these neurons in an ovarian-dependent manner. We isolated and purified these neurons from intact, sham-operated, and ovariectomized females of npba-GFP transgenic medaka[13] and performed RNA sequencing (RNA-seq) (Supplementary Fig. 1a, b). The results for all annotated genes are summarized in Supplementary Data 1. Differential expression analyses using edgeR and cuffdiff (false discovery rate <0.05) identified 171 and 321 differentially expressed genes (DEGs), respectively, both between intact and ovariectomized females and between sham-operated and ovariectomized females (Supplementary Fig. 1c, d). Among these DEGs, 107 showed significant differences in both edgeR and cuffdiff. Of note, these included at least three putative neuropeptide genes (gene ID: XLOC_022840, XLOC_003385, and XLOC_024729), all of which were downregulated by ovariectomy (Fig. 1a). Further annotation revealed that XLOC_022840 is one of six cocaine- and amphetamine-regulated transcript peptide (CARTPT)-like genes identified in medaka (GenBank accession number NM_001204781)[14], and phylogenetic analysis revealed that it encodes the medaka ortholog of zebrafish Cartpt2b (Fig. 1b). XLOC_003385 was found to be the medaka tachykinin 1 gene (tac1) (GenBank accession number AB441191)[15], while XLOC_024729 was identical to a predicted medaka gene in the GenBank database, XM_020705530, which was identified as the medaka ortholog of zebrafish Tac4a by phylogenetic tree analysis (Fig. 1c).

Double in situ hybridization confirmed the expression of these neuropeptide genes in npba-expressing neurons in the PMm/PMg. More specifically, cartpt2b and tac4a were mostly expressed in the anterior (PMm) subpopulation of these neurons, while tac1 was exclusively expressed in the posterior (PMg) subpopulation (Fig. 1d). These results demonstrate that female-specific npba-expressing neurons in the PMm/PMg express a multitude of neuropeptide genes depending on ovarian secretions. Considering this, together with the fact that these neurons occur exclusively in females and are highly dependent on gonadal sex steroids[8,12,13], we termed them female-specific, sex steroid-responsive peptidergic (FeSP) neurons.

### *ptger4b* is expressed in FeSP neurons in an ovarian estrogen-dependent manner.
In addition to the aforementioned neuropeptide genes, the DEGs downregulated by ovariectomy included a putative $PGE_2$ receptor gene, XLOC_004257 (Fig. 2a). Considering that $PGE_2$ has been implicated in the regulation of female sexual receptivity across vertebrate phyla[10], but its neural mechanisms of action remain poorly understood, we selected this gene as the focus of our analysis. Further annotation revealed that XLOC_004257 represents one of the medaka $PGE_2$ receptor 4 (EP4) genes, ptger4b (GenBank accession number NM_001308974)[16], which was confirmed by phylogenetic tree analysis (Fig. 2b). Double in situ hybridization demonstrated that ptger4b is expressed in both the anterior (PMm) and posterior (PMg) subpopulations of FeSP neurons (Fig. 2c). We further validated the ovarian secretion-dependent regulation of ptger4b expression in FeSP neurons by real-time PCR, which was performed on FeSP neurons isolated and purified from females that were sham-operated, ovariectomized, or ovariectomized and treated with estradiol-17β ($E_2$; the major estrogen in vertebrates, including teleosts) or 11-ketotestosterone (KT; the primary, non-aromatizable androgen in teleosts). The results showed that ovariectomy caused a significant decrease in ptger4b expression ($p = 0.0381$), which was recovered by $E_2$ treatment ($p < 0.0001$), whereas KT had no effect ($p > 0.9999$) (Fig. 2d). It can thus be concluded that ovarian estrogens promote the expression of ptger4b in FeSP neurons.

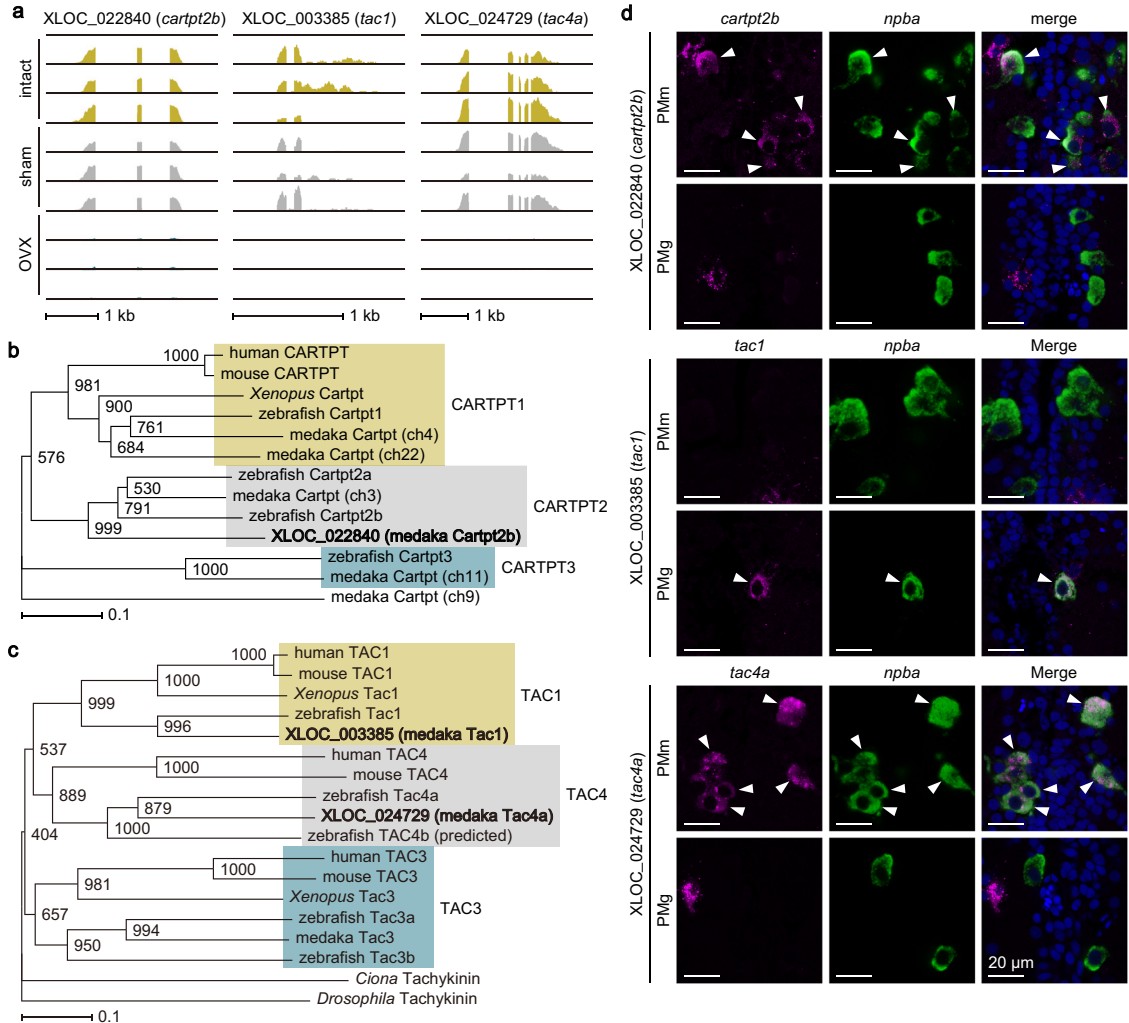

**Fig. 1 Female-specific preoptic neurons express multiple neuropeptides in an ovarian-dependent manner. a** Integrative genomics viewer (IGV) images of RNA-seq reads from female-specific *npba*-expressing neurons in the PMm/PMg of intact, sham-operated (sham), and ovariectomized (OVX) females ($n = 3$ for each treatment) for XLOC_022840 (*cartpt2b*), XLOC_003385 (*tac1*), and XLOC_024729 (*tac4a*). Phylogenetic trees showing the relationship of medaka Cartpt2b (**b**) and Tac1/Tac4 (**c**) to other known CARTPT and tachykinin family proteins. The number at each node indicates bootstrap values for 1000 replicates. Scale bar represents 0.1 substitutions per site. For species names and GenBank accession numbers, see Supplementary Table 1.
**d** Representative micrographs showing the expression of *catrpt2b*, *tac1*, and *tac4* in *npba*-expressing neurons in the PMm (upper panels) and PMg (lower panels). Left and middle panels show images of, respectively, *cartpt2b/tac1/tac4a* (magenta) and *npba* (green) expression in the same section; right panels show the merged images with nuclear counterstaining (blue). Arrowheads indicate representative neurons co-expressing *cartpt2b/tac1/tac4a* and *npba*. Scale bars represent 20 µm.

This finding, along with the fact that FeSP neurons express estrogen receptors[8,11], led us to test the possibility that estrogens directly regulate the transcription of *ptger4b*. The search for potential estrogen-responsive elements (EREs) in the medaka *ptger4b* locus identified two canonical bipartite ERE-like sequences in the 5′-proximal region (at positions −1076 and −714 relative to the transcription start site) (Fig. 2e). Luciferase-based transcriptional activity assays using a 5′-proximal fragment of *ptger4b* containing these two ERE-like sequences revealed that $E_2$ induced a significant increase in luciferase activity in the presence of Esr2a ($p = 0.0056$, 0.0024, 0.0015, and 0.0024 at $10^{-9}$, $10^{-8}$, $10^{-7}$, and $10^{-6}$ M, respectively) and Esr2b ($p < 0.0001$ at all concentrations tested) (Fig. 2f). Although not significant, a slight inhibition was observed in the presence of Esr1 (Fig. 2f).

Next, we introduced point mutations into each of the two canonical bipartite ERE-like sequences in the luciferase reporter construct and examined the resulting change in luciferase activity.

Both mutations of the ERE-like sequences at positions −1076 and −714 abolished $E_2$'s induction of luciferase activity in the presence of Esr2b (Fig. 2g). These results suggest that the stimulatory action of estrogens on *ptger4b* transcription is mediated, at least in part, by Esr2b and that these two EREs act cooperatively with each other. In contrast, the inhibitory and inductive effects of $E_2$ were still observed in the presence of Esr1 and Esr2a, respectively, even with mutations in the ERE-like sequences at positions −1076 ($p = 0.0561$ and 0.0422, respectively) and −714 ($p = 0.0365$ and 0.0129, respectively) (Fig. 2g).

**PGE₂/Ptger4b signaling in the female brain exhibits diurnal fluctuation at the ligand level.** Next, we determined the spatial and temporal patterning of PGE₂/Ptger4b signaling in the male and female brain to provide further insights into its physiological properties. Examination of the distribution of *ptger4b* expression

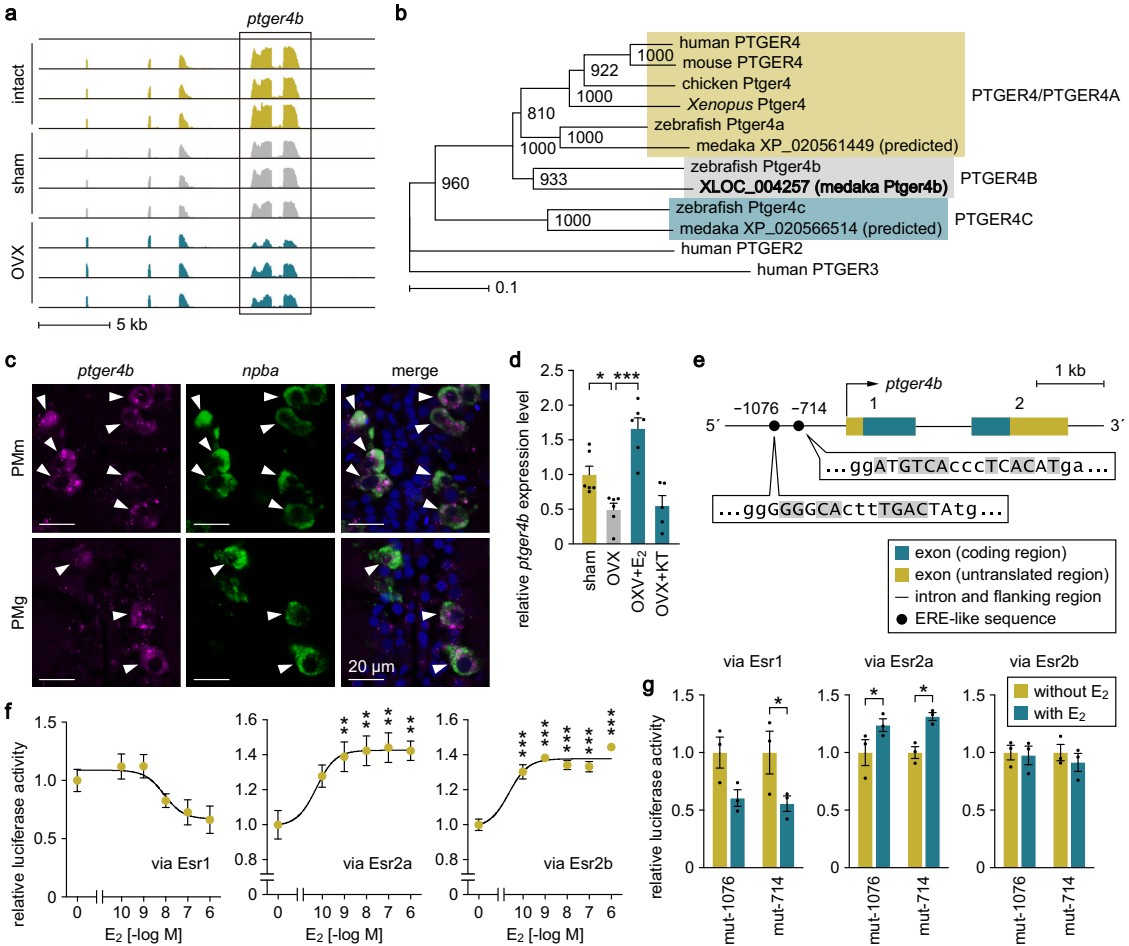

**Fig. 2 *ptger4b* is expressed in FeSP neurons in an ovarian estrogen-dependent manner. a** Integrative genomics viewer (IGV) images of RNA-seq reads from FeSP neurons of intact, sham-operated (sham), and ovariectomized (OVX) females ($n = 3$ for each treatment) for *ptger4b*. **b** Phylogenetic tree showing the relationship of medaka Ptger4b to other known PGE$_2$ receptor 4 proteins. The number at each node indicates bootstrap values for 1000 replicates. Scale bar represents 0.1 substitutions per site. For species names and GenBank accession numbers, see Supplementary Table 1. **c** Representative micrographs showing the expression of *ptger4b* in FeSP neurons in the PMm (upper panels) and PMg (lower panels). Left and middle panels show images of *ptger4b* (magenta) and *npba* (green) expression, respectively, in the same section; right panels show the merged images with nuclear counterstaining (blue). Arrowheads indicate representative FeSP neurons expressing *ptger4b*. Scale bars represent 20 μm. **d** Levels of *ptger4b* expression in FeSP neurons of sham females and OVX females exposed to vehicle only (OVX), estradiol-17β (OVX + E$_2$), or 11-ketotestosterone (OVX + KT) as determined by real-time PCR ($n = 6$ per treatment except OVX + KT, where $n = 5$). Mean value of sham females was arbitrarily set to 1. **e** Genomic structure of *ptger4b* depicting the relative location of the two canonical bipartite estrogen-responsive element (ERE)-like sequences in the 5′-proximal region. Bent arrow indicates the transcription start site. Numbers are shown above each exon. The nucleotide sequence of each ERE-like sequence is shown in uppercase letters. Nucleotides identical to the ERE consensus sequence (AGGTCAnnnTGACCT) are highlighted in gray. **f** Stimulation of *ptger4b* transcriptional activity by E$_2$. CHO cells were transfected with a luciferase reporter construct containing a genomic fragment upstream of the first methionine codon of *ptger4b* and an expression vector for either Esr1, Esr2a, or Esr2b. Cells were treated with different concentrations of E$_2$, and luciferase activity was measured. Values are expressed as fold induction over unstimulated cells. **g** Effects of mutations in ERE-like sequences on the stimulation of *ptger4b* transcriptional activity by E$_2$. Cells were transfected with a luciferase reporter construct carrying a mutation in the ERE-like sequences at position −1076 (mut-1076) or −714 (mut-714), together with the Esr1, Esr2a, or Esr2b expression construct. Cells were treated with or without E$_2$ and luciferase activity was measured. Values are expressed relative to cells without E$_2$ stimulation. Each assay was performed in triplicate and repeated independently three times, except for the dose-response assay with Esr2a, which was repeated six times. Quantitative data were expressed as means with error bars representing standard error of the mean. Statistical differences were assessed by Bonferroni's post hoc test (**d**), Dunnett's post hoc test (versus unstimulated control) (**f**), and unpaired *t*-test (**g**). \**p* < 0.05; \*\**p* < 0.01; \*\*\**p* < 0.001.

throughout the entire brain by in situ hybridization revealed that *ptger4b* was expressed in five brain nuclei: expression in the Vv (ventral nucleus of the ventral telencephalic area) and PPa (anterior parvocellular preoptic nucleus) was common to both sexes; expression in Pbl (basal lateral preoptic nucleus) was male-specific; and expression in the PPp (posterior parvocellular preoptic nucleus) and PMm/PMg, where FeSP neurons reside, was female-specific (Fig. 3a–c).

Given that medaka have a 24 h reproductive cycle (where they spawn at the beginning of the light period), we assessed the diurnal fluctuations of brain *ptger4b* expression and brain levels of PGE$_2$ and its metabolites by real-time PCR and liquid chromatography-tandem mass spectrometry (LC-MS/MS), respectively. In the female brain, there were no significant changes in *ptger4b* expression throughout the day, while in the male brain, *ptger4b* expression was transiently elevated at the

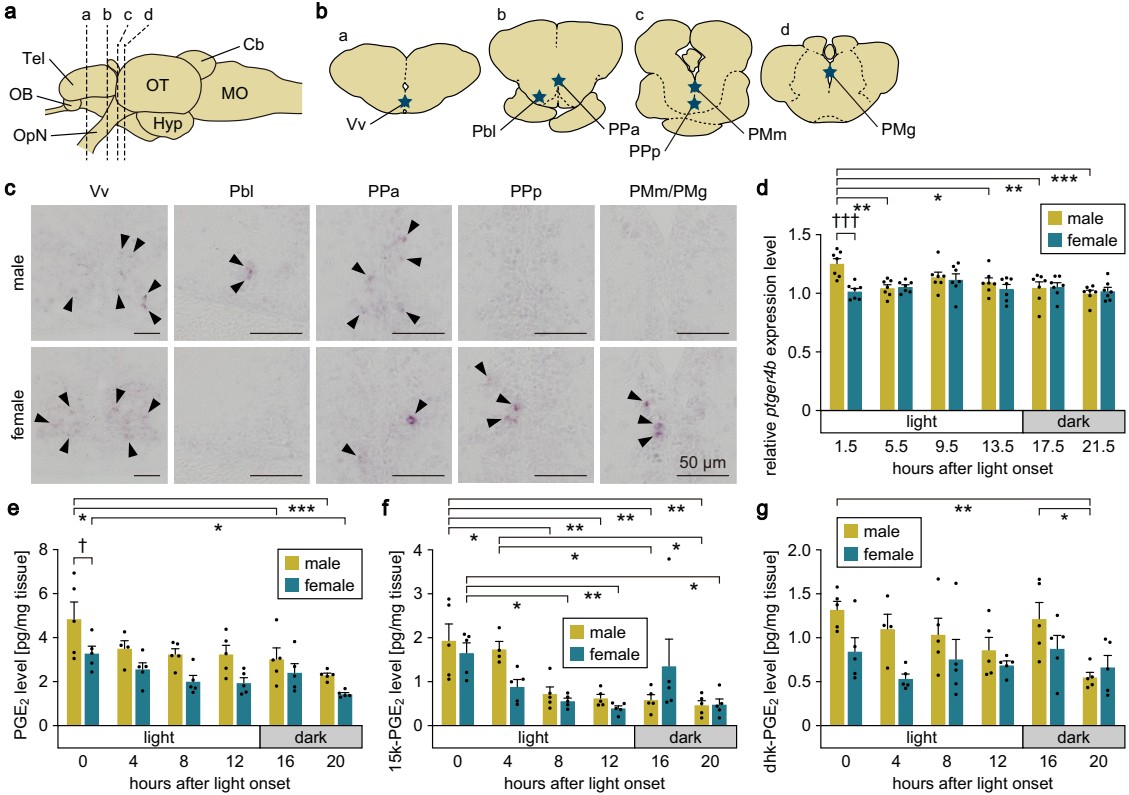

**Fig. 3 $PGE_2$/Ptger4b signaling in the female brain exhibits diurnal fluctuation at the ligand level. a** Lateral view (anterior to the left) of the medaka brain showing the approximate levels of coronal sections presented in panel (**b**). Cb cerebellum; Hyp hypothalamus; MO medulla oblongata; OB olfactory bulb; OpN optic nerve; OT optic tectum; Tel telencephalon. **b** Coronal sections depicting the location of the Vv, Pbl, PPa, PPp, and PMm/PMg. **c** Representative micrographs depicting *ptger4b* expression in the Vv, Pbl, PPa, PPp, and PMm/PMg of males and females. Arrowheads indicate representative *ptger4b*-expressing neurons. Scale bars represent 50 μm. **d** Diurnal fluctuations of *ptger4b* expression in male and female brains as determined by real-time PCR ($n = 7$ per sex and time). Mean value of females at 1.5 h after light onset was arbitrarily set to 1. Diurnal fluctuations of $PGE_2$ (**e**), 15k-$PGE_2$ (**f**), and dhk-$PGE_2$ (**g**) levels in the male and female brains as determined by LC-MS/MS ($n = 5$ per sex and time, except males at 4 h after light onset, where $n = 4$). Quantitative data were expressed as means with error bars representing standard error of the mean. Statistical differences were assessed by Bonferroni's post hoc test (**d–g**). *$p < 0.05$; **$p < 0.01$; ***$p < 0.001$ (between time points of the same sex). †$p < 0.05$; †††$p < 0.001$ (between sexes at the same time point).

beginning of the light period (main effect of time, $p = 0.0051$; main effect of sex, $p = 0.0310$; interaction between time and sex, $p = 0.0050$; $p < 0.0001$ for males versus females at 1.5 h after light onset (halo); $p = 0.0017$, 0.0384, 0.0020, and <0.0001 for 1.5 halo versus 5.5, 13.5, 17.5, and 21.5 halo, respectively, in males) (Fig. 3d). Brain levels of $PGE_2$ were highest at 0 halo and decreased throughout the day to their lowest at 20 halo in both sexes (main effect of time, $p < 0.0001$; main effect of sex, $p < 0.0001$; interaction between time and sex, $p = 0.8229$; $p = 0.0173$ and 0.0002 for 0 versus 16 and 20 halo, respectively, in males; $p = 0.0156$ for 0 versus 20 halo in females) (Fig. 3e). In addition, $PGE_2$ levels were significantly higher in males than in females at 0 halo ($p = 0.0277$), and a similar, though not significant, male bias was noted at other times. Brain levels of 15-keto-$PGE_2$ (15k-$PGE_2$) were lower than $PGE_2$ and followed a similar pattern of fluctuation; they were highest at the beginning of the light period and decreased throughout the day (main effect of time, $p < 0.0001$; main effect of sex, $p = 0.3877$; interaction between time and sex, $p = 0.0669$; $p = 0.0153$, 0.0065, 0.0045, and 0.0015 for 0 versus 8, 12, 16, and 20 halo, respectively, in males; $p = 0.0409$ and 0.0159 for 4 versus 16 and 20 halo, respectively, in males; $p = 0.0385$, 0.0095, and 0.0200 for 0 versus 8, 12, and 20 halo, respectively, in females) (Fig. 3f). No significant difference in 15k-$PGE_2$ levels was found between sexes. 13,14-dihydro-15-keto-$PGE_2$ (dhk-$PGE_2$) levels did not show clear temporal

changes, although a decrease was seen in males at 20 halo ($p = 0.0059$ and 0.0267 versus 0 and 16 halo, respectively) (Fig. 3g).

Taken together, these results indicate that, in the female brain, *ptger4b* expression is fairly constant throughout the day, and instead, $PGE_2$ levels show diurnal fluctuations, peaking rapidly at the beginning of the light period when medaka spawn.

**ptger4b-deficient females retain normal ovarian development and function.** To examine the role of *ptger4b* in mating behavior, we generated *ptger4b* knockout medaka by using clustered regularly interspaced short palindromic repeats (CRISPR)/CRISPR-associated protein 9 (Cas9)-mediated genome editing. Two independent knockout lines (Δ17 and Δ10) were generated for use in this study to ensure the validity of our results and control for possible off-target effects (Fig. 4a). Each line contained a deleterious frameshift mutation, resulting in the early truncation of the Ptger4b protein (Fig. 4b).

Previous evidence suggests that Ptger4b plays a critical role in ovulation in medaka: *ptger4b* expression is highly upregulated in preovulatory follicles, and ovulation can be blocked by an EP4 antagonist[16]. We therefore first examined the effect of *ptger4b* deficiency on ovarian development and function. We found no difference in adult body weight or gonad/body weight ratio

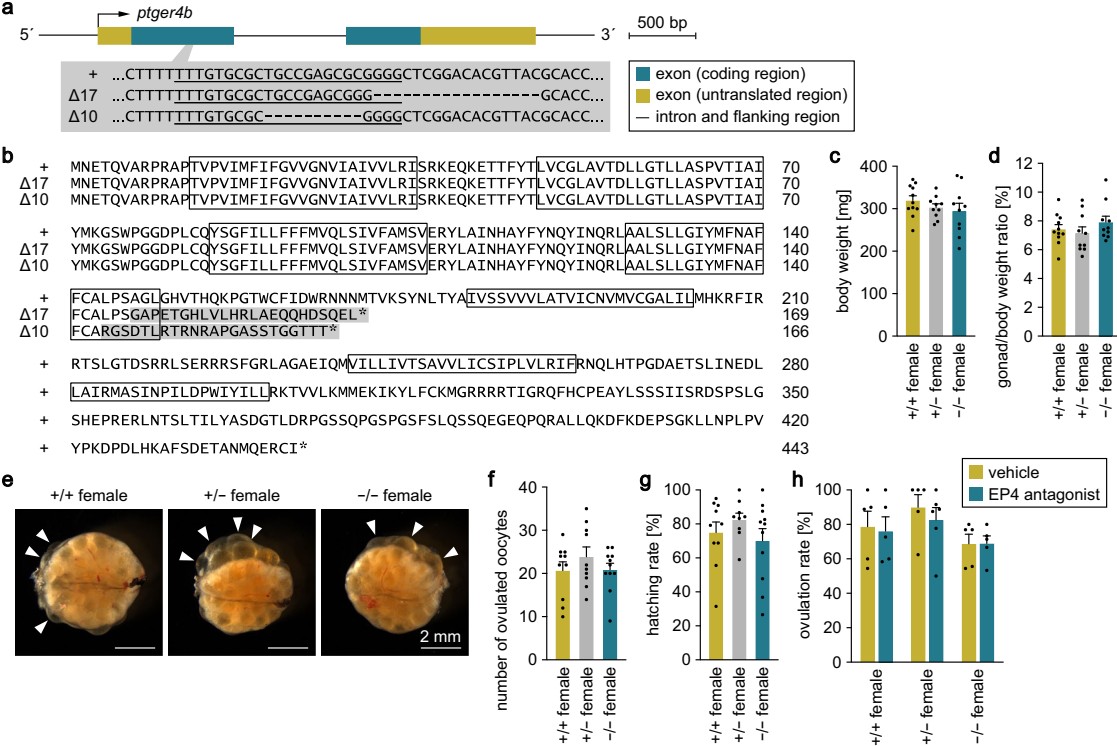

**Fig. 4 *ptger4b*-deficient females retain normal ovarian development and function.** Two independent lines of *ptger4b* knockout medaka (Δ17 and Δ10) were generated by CRISPR/Cas9-mediated genome editing. **a** The gene structure of *ptger4b* is shown with the location of the CRISPR target site, which is enlarged to depict the nucleotide sequences of the wild-type (+) and targeted (Δ17 and Δ10) alleles. The target sequence complementary to the CRISPR RNA is underlined, and deleted nucleotides are indicated by dashes. **b** Comparison of the deduced Ptger4b protein sequences of the +, Δ17, and Δ10 alleles. Boxes represent the predicted transmembrane domains of Ptger4b. The altered sequence caused by frameshift is shaded in gray. Asterisks indicate stop codons. Body weights (**c**) and gonad/body weight ratios (**d**) of adult *ptger4b*$^{+/+}$ ($n = 11$), *ptger4b*$^{+/-}$ ($n = 10$), and *ptger4b*$^{-/-}$ ($n = 10$) females of the Δ17 line. **e** Representative images of the ovaries of adult *ptger4b*$^{+/+}$, *ptger4b*$^{+/-}$, and *ptger4b*$^{-/-}$ females of the Δ17 line. Arrowheads indicate representative ovulated oocytes. Scale bars represent 2 mm. Number of ovulated oocytes (**f**) and hatching rate of fertilized eggs (**g**) of *ptger4b*$^{+/+}$ ($n = 10$), *ptger4b*$^{+/-}$ ($n = 10$), and *ptger4b*$^{-/-}$ ($n = 11$) females of the Δ17 line. **h** Effect of the EP4 antagonist GW 627368X on the ovulation of oocytes from *ptger4b*$^{+/+}$, *ptger4b*$^{+/-}$, and *ptger4b*$^{-/-}$ females in vitro ($n = 5$ for each experimental group except *ptger4b*$^{+/-}$ females treated with the EP4 antagonist, where $n = 6$; each n comprises 8–20 oocytes). Quantitative data were expressed as means with error bars representing standard error of the mean. Statistical differences were assessed by Bonferroni's post hoc test (**c**, **d**, **f–h**).

between genotypes in either Δ17 or Δ10 lines (Fig. 4c, d; Supplementary Fig. 2a, b). There was also no observable difference in gross ovarian morphology between genotypes in both lines (Fig. 4e; Supplementary Fig. 2c). In addition, all genotypes had similar numbers of ovulated oocytes and hatching rate of fertilized eggs (Fig. 4f, g; Supplementary Fig. 2d, e). We further investigated the effect of the EP4 antagonist GW 627368X on ovulation using an in vitro assay as previously reported[16,17] in order to examine the possibility that other EP4 subtypes may compensate for the loss of Ptger4b and help to induce ovulation. The results showed that GW 627368X failed to inhibit ovulation not only in *ptger4b*$^{-/-}$ females, but also in *ptger4b*$^{+/+}$ and *ptger4b*$^{+/-}$ females, in contrast to what was reported previously[16,17] (Fig. 4h). Taken together, these results indicate that *ptger4b* deficiency does not affect ovarian development and function, contrary to expectations based on previous findings.

**_ptger4b_-deficient females exhibit increased sexual receptivity.** Next, we assessed the effect of *ptger4b* deficiency on mating behavior. Medaka mating behavior consists of a stereotyped and quantifiable series of actions that begin with the male approaching and closely following the female[18,19]. The male then performs a courtship display by swimming in a semi-circle in front of the female. If the female is receptive, the male will grasp her with his dorsal and anal fins (termed "wrapping") and they

quiver together ("quivering") until sperm and eggs are released ("spawning") (Fig. 5a). If the female is not receptive, she will either assume a rejection posture or swim rapidly away.

Most females of both Δ17 and Δ10 lines spawned successfully, regardless of genotype (Fig. 5b; Supplementary Fig. 3a). There were no significant differences between genotypes in the number of courtship displays received and wrapping refusals by females of both lines (Fig. 5c, d; Supplementary Fig. 3b, c). However, *ptger4b*$^{-/-}$ females of the Δ17 line exhibited significantly lower latencies from the beginning of the interaction to quivering ($p = 0.0204$), from the first approach to quivering ($p = 0.0180$), and from the first courtship display to quivering ($p < 0.0001$) than *ptger4b*$^{+/+}$ siblings (Fig. 5h–j). Similarly, *ptger4b*$^{-/-}$ females of the Δ10 line showed significantly lower latencies from the first approach to quivering than *ptger4b*$^{+/+}$ ($p = 0.0219$) and *ptger4b*$^{+/-}$ ($p = 0.0162$) siblings and from the first courtship display to quivering than *ptger4b*$^{+/+}$ siblings ($p = 0.0144$), indicating reproducibility between the independent lines (Supplementary Fig. 3h, i). No significant differences were found in the other latency parameters analyzed.

As with females, most males of both Δ17 and Δ10 lines successfully spawned, regardless of genotype (Supplementary Figs. 4 and 5). In males, no significant differences were found between genotypes in either line for any of the behavioral parameters analyzed (Supplementary Figs. 4 and 5). Collectively,

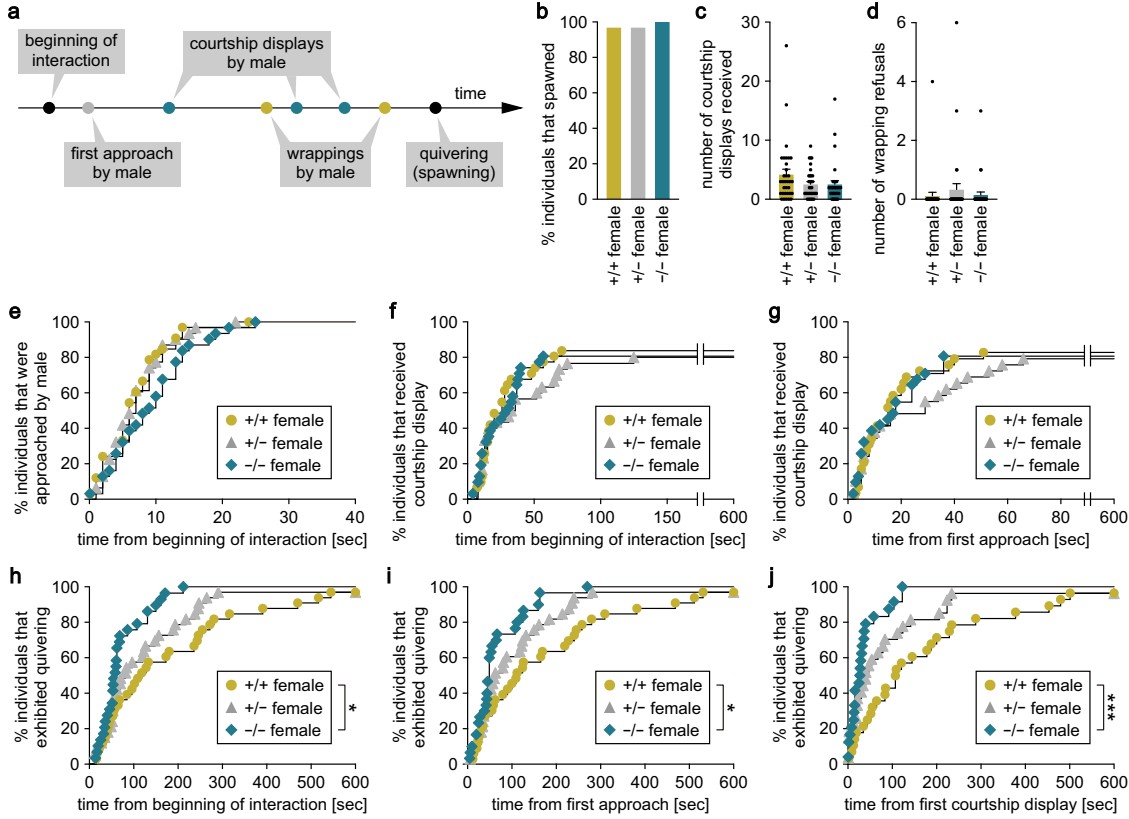

**Fig. 5 *ptger4b*-deficient females exhibit increased sexual receptivity.** *ptger4b*[+/+], *ptger4b*[+/−], and *ptger4b*[−/−] females of the Δ17 line were tested for mating behavior ($n = 33$ for each genotype). **a** Schematic representation of the typical sequence of mating behavior in medaka. **b** Percentage of individuals that spawned during the test period (10 min). Number of courtship displays received (**c**) and wrapping refusals (**d**). Latency from the beginning of interaction to the first approach (**e**) and courtship display (**f**). **g** Latency from the first approach to the first courtship display. Latency from the beginning of interaction (**h**), first approach (**i**), and first courtship display (**j**) to quivering. Quantitative data were expressed as means with error bars representing standard error of the mean (**c, d**). Behavioral time-series data were expressed as Kaplan–Meier plots (**e–j**). Statistical differences were assessed by Fisher's exact test (**b**), Bonferroni's post hoc test (**c, d**), and Gehan–Breslow–Wilcoxon test with Bonferroni's correction (**e–j**). *$p < 0.05$; ***$p < 0.001$.

these results demonstrate that *ptger4b* deficiency has a female-specific effect of increasing sexual receptivity towards male courtship, most likely by shortening the process of perceiving and evaluating male courtship or the subsequent motor response.

**_ptger4b_ deficiency reduces firing activity and impairs neuropeptide release in FeSP neurons.** FeSP neurons are characterized by a large cell body/nucleus and a regular pattern of firing, which are highly dependent on ovarian estrogens[12]. The estrogen-dependent nature of *ptger4b* described above led us to speculate that these cellular phenotypes may be mediated by *ptger4b*. To test this idea, we first analyzed the cell body/nuclear size of FeSP neurons in *ptger4b*-deficient females but found no differences by genotype in either Δ17 or Δ10 lines (Fig. 6a–c; Supplementary Fig. 6a–c). We then analyzed the firing rate of FeSP neurons in *ptger4b*-deficient females by patch-clamp recording. FeSP neurons of *ptger4b*[−/−] females exhibited regular firing patterns similar to *ptger4b*[+/+] siblings, but with significantly lower average firing rates in both Δ17 ($p = 0.0004$) and Δ10 ($p = 0.0366$) lines, indicating a role for *ptger4b* in regulating neuronal firing (Fig. 6d, e; Supplementary Fig. 6d, e).

Next, we hypothesized that the reduced firing rate of FeSP neurons may diminish neuropeptide release in *ptger4b*-deficient females. Immunohistochemical staining for Npba showed that its signal intensity in FeSP neurons of *ptger4b*[−/−] females was significantly higher than *ptger4b*[+/+] siblings ($p < 0.0001$), indicating an intracellular accumulation of Npba (Fig. 6f, g). To exclude the

possibility that this effect was transcriptional, we analyzed *npba* mRNA expression in the PMm/PMg of *ptger4b*-deficient females by in situ hybridization. There was no difference in the total area of *npba* mRNA signal between *ptger4b*[+/+] and *ptger4b*[−/−] females (Fig. 6h, i). Thus, we concluded that *ptger4b* deficiency causes reduced firing activity and impaired neuropeptide release in FeSP neurons.

In rodents, it has been shown that PGE$_2$/EP4 signaling is critical to the establishment of male-typical mating behaviors by sensitizing medial preoptic neurons to glutamatergic sensory afferents[20]. This led us to hypothesize that a similar mechanism may exist in medaka FeSP neurons. To test this hypothesis, we investigated the response of FeSP neurons to glutamatergic stimulation by patch-clamp recordings. We observed a significant increase in the firing rate of FeSP neurons in response to puff glutamate application ($p = 0.0006$) (Supplementary Fig. 7a, b). However, comparison of the magnitude of glutamatergic response of FeSP neurons between *ptger4b*[+/+] and *ptger4b*[−/−] females revealed no significant difference by genotype (Supplementary Fig. 7b, c). These results suggest that *ptger4b* does not regulate the sensitivity of FeSP neurons to glutamatergic inputs. We additionally assessed the effects of PGE$_2$ and the EP4 antagonist GW 627368X on the firing activity of FeSP neurons. FeSP neurons showed no change in firing rate in response to bath application of either PGE$_2$ or GW 627368X (Supplementary Fig. 7d). These results suggest that the effects of PGE$_2$/Ptger4b signaling on FeSP neurons are constitutive rather than acute.

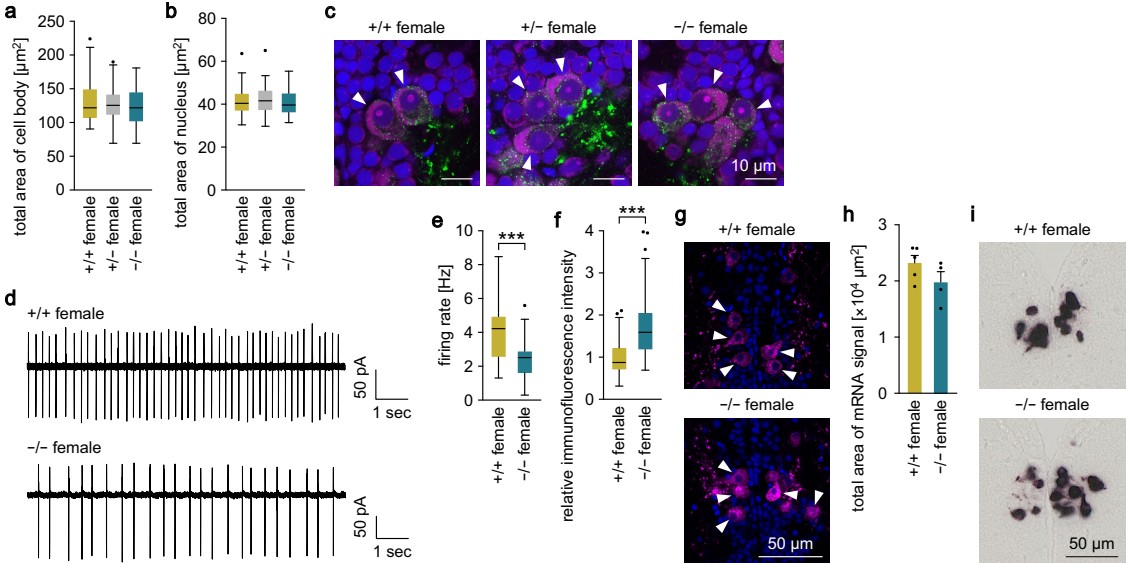

**Fig. 6 *ptger4b* deficiency reduces firing activity and impairs neuropeptide release in FeSP neurons.** Cellular phenotypes of FeSP neurons in the PMm/PMg were analyzed in females of the Δ17 line. Cell body size (**a**) and nuclear size (**b**) of FeSP neurons in *ptger4b*+/+, *ptger4b*+/−, and *ptger4b*−/− females (n = 50 neurons from 5 individuals for each genotype). **c** Representative micrographs showing the morphology of FeSP neurons (arrowheads) in *ptger4b*+/+, *ptger4b*+/−, and *ptger4b*−/− females. Npba immunolabeling is shown in green; Nissl-stained cell bodies in magenta; and DAPI-stained nuclei in blue. Scale bars represent 10 μm. **d** Representative firing patterns of FeSP neurons in *ptger4b*+/+ and *ptger4b*−/− females. **e** Firing frequencies of FeSP neurons in *ptger4b*+/+ (n = 21 neurons from 10 individuals) and *ptger4b*−/− (n = 25 neurons from 14 individuals) females. **f** Relative immunofluorescence intensity of Npba in FeSP neurons of *ptger4b*+/+ (n = 54 neurons from three individuals) and *ptger4b*−/− (n = 98 neurons from three individuals) females. **g** Representative micrographs showing Npba immunolabeling (magenta) in FeSP neurons (arrowheads) of *ptger4b*+/+ and *ptger4b*−/− females. DAPI-stained nuclei are shown in blue. Scale bars represent 50 μm. **h** Total area of *npba* mRNA signal in the PMm/PMg of *ptger4b*+/+ (n = 5) and *ptger4b*−/− (n = 4) females. **i** Representative micrographs showing *npba* mRNA signal in the PMm/PMg of *ptger4b*+/+ and *ptger4b*−/− females. Scale bars represent 50 μm. Quantitative data were expressed as box-and-whisker plots (**a**, **b**, **e**, **f**) or means with error bars representing standard error of the mean (**h**). Statistical differences were assessed by Bonferroni's post hoc test (**a**, **b**), unpaired *t*-test (**e**, **h**), and unpaired *t*-test with Welch's correction (**f**). ***$p < 0.001$.

## Discussion

Here, we show by transcriptomic analysis that female-specific *npba*-expressing neurons in the medaka PMm/PMg express a multitude of neuropeptides, including *cartpt2b*, *tac1*, and *tac4a*, in an ovarian-dependent manner, and we thus termed them FeSP neurons. A recent single-cell transcriptomic study in the mouse preoptic area revealed that many populations of behaviorally relevant neurons co-express multiple neuropeptides[21]. Together with this study, our findings suggest the evolutionary conservation of the multipeptidergic nature of preoptic neurons. We previously showed that *npba* (most likely *npba* expressed in FeSP neurons) plays a critical role in female sexual receptivity[13]. It remains to be determined whether other neuropeptides expressed in FeSP neurons are also relevant to female receptivity; however, their marked ovarian dependence, as well as evidence available in other species, suggests that this may be the case. For example, administration of substance P (encoded by *Tac1*) into the female rat brain facilitates lordosis, a stereotyped mating posture adopted by sexually receptive female rodents, and substance P receptor-deficient female mice display a decreased preference for male sexual pheromones[22,23]. In addition, knockdown of natalisin, an insect tachykinin-like peptide, suppresses mating behavior in female *Drosophila*[24]. Collectively, FeSP neurons may integrate ovarian signaling and multiple different neuropeptide signaling pathways to regulate female receptivity.

This idea is further corroborated by the finding that FeSP neurons express *ptger4b* in an ovarian estrogen-dependent manner, as PGE₂ signaling has been implicated in female receptivity across vertebrate phyla. Similar induction of EP4 expression by estrogens has been reported in the smooth muscle of the ewe cervix and bovine oviduct[25,26]. Additionally, in female

rats, PGE₂ affects the excitability of preoptic gonadotropin-releasing hormone 1 (GnRH1) neurons via EP4, depending on the presence of estrogens[27], suggesting that the expression of EP4 in these neurons is dependent on estrogens, as is the case in FeSP neurons. Therefore, the enhancement of PGE₂/EP4 signaling by estrogens may be a conserved mechanism across species. We also found that the estrogenic induction of *ptger4b* expression in FeSP neurons is mediated, at least in part, by direct transcriptional activation by Esr2b (which is the main estrogen receptor expressed in FeSP neurons) and that deficiency of *ptger4b* alters female receptivity. Given that *ptger4b*-deficient females had no abnormalities in ovarian function and males did not show behavioral defects, the observed alteration in female receptivity is probably due to the loss of female-specific neural *ptger4b* expression (most likely in PMm/PMg FeSP neurons). Taken together, PGE₂/Ptger4b signaling may act downstream of E₂/Esr2b signaling in FeSP neurons to regulate female receptivity. While studies in rats have proposed that the site of action of PGE₂ on female receptivity is preoptic GnRH1 neurons[28], our results strongly suggest that FeSP neurons are a major target of PGE₂ in medaka via Ptger4b. Of note, in medaka, *ptger4b* expression was evident in males but not in females in the Pbl, where GnRH1 neurons reside. The varying effect of PGE₂ on female receptivity between species may reflect differences in its site of action within behaviorally relevant circuits.

While females deficient for *esr2b* were completely unreceptive to male courtship[8], deficiency of *npba* and *ptger4b*, which act downstream of E₂/Esr2b signaling, had much milder impacts on female receptivity[13]. A plausible explanation for this may be that E₂/Esr2b signaling controls most, if not all, components of female receptivity by simultaneously regulating multiple behaviorally

relevant genes, including *npba* and *ptger4b*, and that each of these genes is separately responsible for one or a few components. This assumption is compatible with the view that sexually dimorphic social behaviors are regulated in a modular manner by multiple sexually dimorphic genes acting downstream of sex steroid signaling[29]. It should be noted that, in contrast to *npba* deficiency, which led to increased latency to quivering/spawning, *ptger4b* deficiency decreased spawning latency, suggesting that *npba* and *ptger4b* have an overall facilitatory and inhibitory effect on female receptivity, respectively. According to their multipeptidergic nature, FeSP neurons may simultaneously activate multiple signaling pathways with opposite effects to fine-tune female receptivity. Facilitatory and inhibitory signaling within single peptidergic neurons has also been demonstrated in kisspeptin/neurokinin B/dynorphin neurons in the arcuate nucleus of mammals (where kisspeptin and neurokinin B activate, whereas dynorphin inhibits, the pulsatile release of GnRH1)[30]. Although further studies are clearly needed, it seems plausible that such a regulatory system is a more general feature of peptidergic neurons than has been recognized.

We further found that *ptger4b* deficiency reduced the firing activity of FeSP neurons, suggesting a regulatory effect of $PGE_2$/Ptger4b on the electrophysiological properties of these neurons. Our previous data showed that the firing of FeSP neurons is dependent on ovarian estrogens and that these neurons do not show diurnal changes in firing rate[12]. Considering that *ptger4b* expression in FeSP neurons requires ovarian estrogens, it is probable that the estrogen-dependent electrophysiological activity of these neurons is mediated, at least in part, by $PGE_2$/Ptger4b signaling. Interestingly, given that the firing rates of FeSP neurons are not altered by acute treatment with $PGE_2$/EP4 antagonist and do not correspond to the diurnal fluctuation of brain $PGE_2$ levels, the effects of $PGE_2$/Ptger4b signaling on the electrophysiological properties of FeSP neurons are seemingly constitutive, rather than acute and transient. These effects are in contrast to those in mammalian peptidergic neurons, where $PGE_2$/EP4 signaling typically mediates acute changes in excitability[27,31].

The question then arises as to how the firing activity of FeSP neurons affects female receptivity. We observed increased Npba immunolabeling in FeSP neurons of *ptger4b*-deficient females with no accompanying change in *npba* mRNA levels, suggesting that loss of *ptger4b* expression impairs Npba release in these neurons. It thus seems highly likely that increased firing by $PGE_2$/Ptger4b signaling serves to potentiate the release of neuropeptides. Generally, the exocytosis of neuropeptides requires high-frequency burst firing; it is therefore probable that the firing activity of FeSP neurons serves an important priming function, facilitating the release of neuropeptides in response to certain excitatory cues. The observed impaired Npba release (and the consequent likely reduction in Npba signaling) in *ptger4b*-deficient females is seemingly inconsistent with the opposite behavioral phenotypes observed in *ptger4b*- and *npba*-deficient females. As revealed by transcriptomic analysis, however, FeSP neurons are multipeptidergic, expressing many neuropeptides in addition to Npba. Peptidergic neurons also often co-produce classical small-molecule neurotransmitters, such as glutamate and γ-aminobutyric acid (GABA)[32]. Hence, the behavioral phenotype observed in *ptger4b*-deficient females may be due to an overall reduction in neuropeptides, and possibly also small-molecule neurotransmitters, released from FeSP neurons and not exclusively due to reduced Npba signaling.

Notably, $PGE_2$/EP4 signaling has been shown to be critical for the induction of male-typical synaptic patterning in the developing medial preoptic neurons of rodents and the consequent manifestation of male-typical mating behavior in adulthood[20]. This mechanism involves $PGE_2$-induced phosphorylation and

mobilization of AMPA-type glutamate receptors to the cell membrane of preoptic neurons and their resulting sensitization to glutamatergic inputs[20]. Based on these pieces of evidence, we hypothesized that a similar system may exist in medaka FeSP neurons, involving enhanced sensitivity to glutamate by $PGE_2$/Ptger4b signaling. However, our results showed that *ptger4b* deficiency did not alter the response of FeSP neurons to glutamate. Given that the effects of $PGE_2$/EP4 signaling in medaka FeSP neurons appear to be constitutive, as described above, unlike those reported in mammals, the mode of action of $PGE_2$/EP4 signaling may differ between medial preoptic neurons in rodents and FeSP neurons in medaka.

Our study adds to the prevailing view that prostaglandins are major regulators of female receptivity in teleosts, demonstrating a role for $PGE_2$ in addition to $PGF_{2\alpha}$. However, these two prostaglandin species differ greatly in their mode of action. $PGF_{2\alpha}$ is essential for female receptivity and is derived from the ovary or oviduct, where the presence of recently ovulated eggs triggers the production and release of $PGF_{2\alpha}$, which then acts as a hormonal signal of reproductive status to the brain to induce mating behaviors[1,9,33,34]. In contrast, $PGE_2$ levels in the ovary do not show diurnal changes corresponding to reproductive state[16], meaning that the diurnal fluctuations of $PGE_2$ and its metabolites in the brain—which are highest at the beginning of the light period when females spawn—are not derived from the ovary. Considering this and data from zebrafish (*Danio rerio*), which showed $PGE_2$ had no effect on mating behavior when administered to the water[35], it is probable that the import of systemic $PGE_2$ into the brain is limited, and that the brain is the primary source of behaviorally relevant $PGE_2$. Thus, $PGE_2$ most likely acts as a locally synthesized neuromodulator in the brain and inhibits female receptivity when binding to Ptger4b. It is not currently clear what the functional significance of the inhibitory nature of $PGE_2$/Ptger4b signaling is to female mating behavior. It may be that $PGE_2$/Ptger4b delays spawning to allow for the careful assessment of male suitability/mate choice, but further work is required to validate this idea.

In summary, here we found that female-specific Npba neurons located in a preoptic nucleus homologous to the PVN are multipeptidergic, expressing a multitude of neuropeptides in an ovarian-dependent manner. We also found that $PGE_2$/Ptger4b signaling in these neurons (termed FeSP neurons) is activated by ovarian estrogens and that $PGE_2$/Ptger4b signaling is inhibitory to female sexual receptivity in medaka. Together, these lines of evidence suggest that FeSP neurons act as a relay point, converting ovarian estrogen signals into various inter- (neuropeptides) and intra- ($PGE_2$/Ptger4b) neural signaling molecules which, presumably, affect female receptivity. Although neurons that correspond to FeSP neurons have not been identified in other species, both NPB and EP4 have been shown to be expressed in the rat PVN[36,37]. It would therefore be worthwhile to investigate whether the findings in medaka are applicable to other vertebrates, including rodents. Given that the effects of $PGE_2$ on female receptivity vary greatly among species with facilitative effects in many species, including rats and hamsters, and inhibitory effects in guinea pigs and anole lizards[10], the regulatory mechanism of female receptivity by $PGE_2$ found in medaka may be shared only with some species. Additionally, although $PGE_2$/EP4 signaling has been shown to be essential for the establishment of male-typical mating behaviors in rodents[20], no abnormalities were observed in the mating behavior of *ptger4b*-deficient male medaka. This suggests that the effects of $PGE_2$/EP4 signaling on male mating behavior may also differ among species. Our findings provide a possible mechanism by which $PGE_2$ exerts different effects on the evolutionarily conserved behavioral circuitry in different species. However, the findings in

medaka alone do not fully explain the divergent effects of PGE$_2$, and the significance of the inhibitory effects of PGE$_2$/Ptger4b in mature females capable of spawning remains unclear. Further comparative studies on species with different reproductive strategies are needed to address these issues.

## Methods

**Animals**. Wild-type medaka of the d-rR strain and *npba*-GFP transgenic medaka, which express GFP under the control of regulatory regions of *npba*[13], were raised at 28 °C with a 14 h light/10 h dark photoperiod. Expression of GFP in *npba*-GFP transgenic fish is specifically localized in Npba-expressing neurons[13]. Fish were fed 3 to 4 times per day with live brine shrimp and commercial pellet food (Otohime; Marubeni Nisshin Feed, Tokyo, Japan). Sexually mature, spawning adults (aged 3 to 5 months) were used in all experiments and assigned randomly to experimental groups. Fish were sampled at 1 to 3 halo in all analyses except for the diurnal measurements of brain *ptger4b* expression and brain levels of PGE$_2$ and its metabolites, where fish were sampled 6 times a day at 4 h intervals.

All animal procedures were performed in accordance with the guidelines of the Institutional Animal Care and Use Committee of the University of Tokyo. The committee requests the submission of an animal-use protocol only for use of mammals, birds, and reptiles, in accordance with the Fundamental Guidelines for Proper Conduct of Animal Experiment and Related Activities in Academic Research Institutions under the jurisdiction of the Ministry of Education, Culture, Sports, Science and Technology of Japan (Ministry of Education, Culture, Sports, Science and Technology, Notice No. 71; June 1, 2006). Accordingly, we did not submit an animal-use protocol for this study, which used only teleost fish and thus did not require approval by the committee.

**Ovariectomy and drug treatments**. The ovary was removed from each *npba*-GFP transgenic female[13] under tricaine methane sulfonate anesthesia (0.02%) through a small incision in the ventrolateral abdominal wall. After removal of the ovary, the incision was sutured with nylon thread. Sham-operated females underwent the same operation except for removal of the ovary. For RNA-seq, these females were sampled 9 days after the operation. For real-time PCR analysis, ovariectomized females were immersed in water containing 100 ng ml$^{-1}$ of E$_2$, KT, or vehicle (ethanol) only for 6 days after a 3-day recovering period. Sham-operated females were treated with only vehicle and used as controls. The sex steroid concentrations used were based on previously reported serum steroid levels in medaka[38].

**Isolation and purification of FeSP neurons**. FeSP neurons were isolated and purified from intact, sham-operated, and ovariectomized *npba*-GFP transgenic females[13] according to the method of Abe and Oka[39]. In brief, the brain was dissected out and placed in a Petri dish containing Leibovitz's L-15 medium. The PMm/PMg region containing GFP-labeled FeSP neurons was microdissected with fine metal needles (MicroChisel; Eppendorf, Hamburg, Germany) under a fluorescent stereomicroscope. The dissected piece of brain was incubated in Hank's balanced salt solution containing 10 U ml$^{-1}$ papain, 0.8 mM ethylene glycol tetraacetic acid, and 20 mM glucose for 30 min at 28 °C. After transferring to Leibovitz's L-15 medium containing 20 U ml$^{-1}$ DNase I and 5% fetal bovine serum (FBS), the samples were gently triturated using approximately 470-, 300-, 220-, 150-, and 120 μm diameter fire-polished Pasteur pipettes. Dissociated cells were resuspended in Leibovitz's L-15 medium containing 5% FBS and poured into a Petri dish. Individual GFP-labeled FeSP neurons were handpicked with a glass pipette mounted on a micromanipulator (Narishige, Tokyo, Japan) under a fluorescence stereomicroscope M165FC (Leica Microsystems, Wetzlar, Germany).

**RNA-seq**. RNA-seq was performed on FeSP neurons isolated and purified from intact, sham-operated, and ovariectomized females. RNA extraction, cDNA synthesis, and subsequent amplification were performed as described elsewhere[40]. In brief, FeSP neurons (50 neurons per sample, $n = 3$ for each experimental group) were lysed with Buffer RLT (Qiagen, Hilgen, Germany) containing 1% β-mercaptoethanol. Total RNA was purified with Agencourt AMPure XP beads (Beckman Coulter, Brea, CA) and reverse transcribed using SuperScript III reverse transcriptase (Thermo Fisher Scientific, Waltham, MA) and the oligo(dT) primer (5′-TATAGAATTCGCGGCCGCTCGCGATAATACGACTCACTA-TAGGGCG(T)$_{24}$-3′). The resulting first-strand cDNA was purified with Agencourt AMPure XP beads (Beckman Coulter) and treated with exonuclease I (Takara Bio, Shiga, Japan) for primer digestion. After the addition of a poly(A) tail with terminal deoxynucleotidyl transferase (Roche Diagnostics, Basel, Switzerland), second-strand cDNA was synthesized using MightyAmp DNA polymerase (Takara Bio) and the tagging primer (5′-TATAGAATTCGCGGCCGCTCGCGA(T)$_{24}$-3′). The cDNA was then amplified by suppression PCR (18 cycles of 98 °C for 10 s, 65 °C for 15 s, and 68 °C for 5 min) using MightyAmp DNA polymerase (Takara Bio) and the 5′-aminated primer (5′-NH$_2$-GTATAGAATTCGCGGCCGCTCGCGAT-3′) and purified using MinElute PCR Purification Kit (Qiagen).

Sequencing libraries were prepared using NEBNext Ultra DNA Library Prep Kit for Illumina and NEBNext Multiplex Oligo for Illumina (New England Biolabs, Ipswich, MA) and sequenced with single-end 51 bp reads on a Hiseq 1500 System

(Illumina, San Diego, CA). 39–61 million sequences were obtained from each sample (Supplementary Fig. 1b). Sequences were aligned to the reference medaka genome (oryLat2 assembly)[41] using TopHat (ver. 2.0.13)[42] and subsequently Subjunc[43]. Cufflinks (ver. 2.2.1)[44] was then used to assemble the transcripts and estimate their abundance. Pairwise differential expression analysis between samples was performed using the edgeR[45] and cuffdiff (ver. 2.2.1)[46] packages with false discovery rate <0.05. Raw mapped reads were visualized and manually inspected using Integrative Genomics Viewer (IGV)[47].

**In silico cloning and phylogenetic tree analysis**. BLAST searches using the sequences obtained by RNA-seq (gene ID: XLOC_022840, XLOC_003385, XLOC_024729, and XLOC_004257) as queries were performed to identify the corresponding cDNAs in the GenBank nucleotide database, whose sequences were further verified by identification of the corresponding expressed sequence tag (EST) clones in the medaka EST database at National BioResource Project (NBRP) Medaka (http://www.shigen.nig.ac.jp/medaka/). Transmembrane domains of Ptger4b were predicted using the TMHMM server (ver. 2.0; https://services.healthtech.dtu.dk). The deduced amino acid sequences of the identified cDNAs from medaka were aligned with those of their putative orthologs/paralogs from different species by using ClustalW. The resulting alignments were used to construct bootstrapped (1000 replicates) neighbor-joining trees (http://clustalw.ddbj.nig.ac.jp/index.php). Medaka Cartpt (ch9), *Drosophila* tachykinin, and human PTGER3 were used as outgroups to root the trees of CARTPT, tachykinin, and PTGER4, respectively. The species names and GenBank accession numbers of the sequences used are listed in Supplementary Table 1.

**Double-label in situ hybridization**. A 645 bp DNA fragment corresponding to nucleotides 1–645 of the medaka *npba* cDNA (NM_001308979) was PCR-amplified and transcribed in vitro to generate a fluorescein-labeled cRNA probe using Fluorescein RNA Labeling Mix and T7 RNA polymerase (Roche Diagnostics). DNA fragments corresponding to nucleotides 1–532 (532 bp) of the *cartpt2b* cDNA (GenBank accession number NM_001204781), 2–393 (392 bp) of the *tac1* cDNA (DK021817), 100–814 (715 bp) of the *tac4a* cDNA (XM_020705530), and 556–1984 (1429 bp) of the *ptger4b* cDNA (NM_001308974) were PCR-amplified and transcribed to generate digoxigenin (DIG)-labeled cRNA probes using DIG RNA Labeling Mix and T7 RNA polymerase (Roche Diagnostics).

The procedure for double-label in situ hybridization has been described previously[48]. In brief, whole brains removed from females were fixed in 4% paraformaldehyde (PFA) and embedded in 5% agarose supplemented with 20% sucrose. Brains were cut into 20 μm frozen sections in the coronal plane and hybridized simultaneously with the DIG-labeled *cartpt2b*, *tac1*, *tac4a*, or *ptger4b* probe and fluorescein-labeled *npba* probe. The fluorescein-labeled probe was visualized using a horseradish peroxidase-conjugated anti-fluorescein antibody (RRID: AB_2737388; PerkinElmer, Waltham, MA) and the TSA Plus Fluorescein System (PerkinElmer); the DIG-labeled probes were visualized using an alkaline phosphatase-conjugated anti-DIG antibody (RRID: AB_514497; Roche Diagnostics) and Fast Red (Roche Diagnostics). Cell nuclei were counterstained with 4′,6-diamidino-2-phenylindole (DAPI). Fluorescent images were acquired with a confocal laser-scanning microscope (TCS SP8; Leica Microsystems), using excitation and emission wavelengths of 405 nm and 410–480 nm for DAPI, 488 nm and 495–545 nm for fluorescein, and 552 nm and 620–700 nm for Fast Red.

**Real-time PCR**. For analysis of *ptger4b* expression in FeSP neurons, GFP-expressing cells in the PMm/PMg were isolated and purified from sham-operated, ovariectomized, ovariectomized plus E$_2$-treated, and ovariectomized plus KT-treated *npba*-GFP females as described above. RNA extraction, reverse transcription, and cDNA amplification were performed as in RNA-seq. For analysis of diurnal fluctuations in brain *ptger4b* expression, whole brains were sampled from females every 4 h over a 24 h period, beginning at 1.5 halo. Total RNA was extracted from the brains using the RNeasy Lipid Tissue Mini Kit with DNase treatment (Qiagen), and cDNA was synthesized using the Omniscript RT Kit (Qiagen).

Real-time PCR was performed on the LightCycler 480 System II using the LightCycler 480 SYBR Green I Master (Roche Diagnostics). Melt curve analysis was performed to verify that a single amplicon was obtained in each sample. Data from whole brain samples were normalized to the β-actin gene (*actb*; GenBank accession number NM_001104808). Data from FeSP neurons were normalized to the geometric mean of *actb* and the elongation factor 1 α gene (*eef1a*; NM_001104662), according to Vandesompele et al.[49]. The primers used for real-time PCR are listed in Supplementary Table 2.

**Transcriptional activity assay**. A fosmid clone (golwfno354_c17) containing the medaka *ptger4b* locus was obtained from NBRP Medaka. The 5′-proximal region of *ptger4b* (4.0 kb) was analyzed for the presence of potential EREs by using Jaspar (ver. 5.0_alpha) with default settings. A 4675 bp fragment of genomic DNA containing 4422 bp of the 5′-proximal region and the entire 5′-untranslated region was PCR-amplified from the fosmid clone and inserted into the NheI site of the pGL4.10 luciferase reporter vector (Promega, Madison, WI). The resulting

luciferase reporter construct was transiently transfected into CHO cells (obtained from and authenticated by Riken BRC Cell Bank) with an expression construct for either medaka Esr1, Esr2a, or Esr2b[8] and an internal control vector pGL4.74 (Promega) using Lipofectamine LTX (Thermo Fisher Scientific). Cells were treated with $E_2$ in phenol red-free Dulbecco's modified Eagle's medium supplemented with 5% charcoal/dextran-treated FBS (Cytiva, Marlborough, MA) at concentrations of 0, $10^{-10}$, $10^{-9}$, $10^{-8}$, $10^{-7}$, and $10^{-6}$ M for 18 h. The luciferase activity of cell lysates was measured on the GloMax 20/20n Luminometer (Promega) using the Dual-Luciferase Reporter Assay System (Promega). Each assay was performed in triplicate and repeated independently three times, except for the assay with Esr2a, which was repeated six times.

To determine the EREs responsible for the estrogenic induction of *ptger4b* transcription, both half-sites of each of the two potential EREs (at positions −1076 and −714 relative to the transcription start site) were mutated into a HindIII site (AAGCTT) using the PrimeSTAR Mutagenesis Basal Kit (Takara Bio). Assays with these mutated constructs were performed as described above, except that a single dose of $E_2$ ($10^{-6}$ M) was used.

**Single-label in situ hybridization**. Single-label in situ hybridization was performed as described previously[48]. Briefly, whole brains were fixed in 4% PFA and embedded in paraffin. Serial coronal sections of 10 μm thickness were cut and hybridized with the above-mentioned *ptger4b* or *npba* probe, which was labeled with DIG using DIG RNA Labeling Mix and T7 RNA polymerase (Roche Diagnostics). Hybridization signals were visualized using an alkaline phosphatase-conjugated anti-DIG antibody (RRID: AB_514497; Roche Diagnostics) and 5-bromo-4-chloro-3-indolyl phosphate/nitro blue tetrazolium (BCIP/NBT) substrate (Roche Diagnostics). Color development was allowed to proceed overnight (for analysis of *ptger4b* expression) or was stopped within 15 min to avoid saturation (for analysis of *npba* expression). Brain nuclei were identified using the medaka brain atlas[50]. For quantification of *npba* expression in FeSP neurons, images were acquired with a virtual slide microscope (VS120; Olympus, Tokyo, Japan) and the total area of *npba* expression signal in the PMm/PMg was calculated using Olyvia software (Olympus).

**LC-MS/MS**. Whole brains were sampled from both sexes every 4 h over a 24 h period, beginning at 0 halo (the brains from 3 fish were pooled per sample), rapidly frozen, and stored at −80 °C until analysis. Males and females sampled at 0 halo were separated by a perforated, transparent partition to prevent spawning. At all other time points, males and females were allowed to interact freely.

Brain levels of $PGE_2$ and its metabolites, $15k\text{-}PGE_2$ and $dhk\text{-}PGE_2$, were measured by LC-MS/MS essentially as described elsewhere[51]. In brief, the brain samples were homogenized with methanol and solid phase extraction was performed using Oasis HLB extraction cartridges (Waters Corporation, Milford, MA). Chromatography was performed with a Nexera X2 high-performance liquid chromatography system (Shimadzu, Kyoto, Japan) connected to a QTRAP 5500 triple quadrupole mass spectrometer (Sciex, Framingham, MA). Analytes were separated on a Kinetex C18 reverse phase column (150 × 2.1 mm, 1.7 μm) (Phenomenex, Torrance, CA). Mobile phase A consisted of water with 0.1% formic acid, and mobile phase B of acetonitrile. The following stepwise gradient was applied: 30% (0–1 min), 80% (5–6 min), 100% (8–9.5 min), and 30% (9.51–12 min). The flow rate was set to 0.3 ml min$^{-1}$. Electrospray ionization in negative ion mode and multiple reaction monitoring (MRM) were used to achieve high specificity and sensitivity for the simultaneous detection of prostaglandins. The MRM transitions of $PGE_2$, $15k\text{-}PGE_2$, and $dhk\text{-}PGE_2$ were $m/z$ 351 > 271, $m/z$ 349 > 235, and $m/z$ 351 > 175, respectively. The amount of each analyte was calculated by reference to the standard curve and corrected by the percent recovery of the deuterium-labeled internal standard ($PGE_2$-d4).

**Generation of knockout medaka**. Knockout medaka for *ptger4b* were generated by CRISPR/Cas9-mediated genome editing. A CRISPR RNA (crRNA) was designed to target the fourth predicted transmembrane domain (Fig. 4a, b). crRNA and trans-activating CRISPR RNA (tracrRNA) were synthesized by Fasmac (Kanagawa, Japan). Cas9 mRNA was synthesized by in vitro transcription of the linearized pCS2+hSpCas9 plasmid (Addgene plasmid number 51815; Addgene, Cambridge, MA) using the mMessage mMachine SP6 Kit (Thermo Fisher Scientific). crRNA, tracrRNA, and Cas9 mRNA were co-microinjected into the cytoplasm of embryos at the one-cell stage. Potential founders were screened by outcrossing with wild-type fish and testing progeny for mutations at the target site using T7 endonuclease I assay followed by direct sequencing. Two founders were selected that yielded progeny carrying deleterious frameshifts leading to premature truncation of the Ptger4b protein: the progeny of one founder carried a 19 bp deletion and 2 bp insertion (Δ17) and progeny of the other carried a 10 bp deletion (Δ10). These progeny were intercrossed to establish knockout lines (Δ17 and Δ10 lines). Each line was maintained by breeding *ptger4b*$^{+/-}$ males and females to obtain *ptger4b*$^{+/+}$, *ptger4b*$^{+/-}$ and *ptger4b*$^{-/-}$ siblings for experimental use. The genotype of each fish was determined by direct sequencing and high-resolution melt analysis using the primers listed in Supplementary Table 2. In each subsequent experiment, siblings of different genotypes of the same knockout line

raised under the same conditions were used as a comparison group to control for genetic and environmental factors.

**In vitro ovulation assay**. An in vitro ovulation assay was performed according to the method of Fujimori et al.[16,17]. In brief, ovaries were removed from *ptger4b*$^{+/+}$, *ptger4b*$^{+/-}$, and *ptger4b*$^{-/-}$ females of the Δ17 knockout line at 12 halo. All large-sized ovarian follicles were isolated from each ovary, split evenly into control and treatment groups, and incubated in 90% Medium 199 with Earle's salts (Thermo Fisher Scientific) containing 50 mg ml$^{-1}$ gentamycin and either vehicle (dimethyl sulfoxide) alone or 20 μM of the EP4 antagonist GW 627368X (Cayman Chemical Company, Ann Arbor, MI), respectively. The number of oocytes that successfully ovulated was counted the following day at 5 halo. The ovulation rate was presented as the percentage of oocytes that ovulated.

**Mating behavior test**. The mating behavior tests were performed as described elsewhere[13]. In brief, on the day before behavioral testing, each focal male/female (from Δ17 and Δ10 knockout lines) was paired with a stimulus fish of the opposite sex (wild-type d-rR strain) in a 2-litre tank and separated by a perforated, transparent partition. The partition was removed at 1 halo and fish were allowed to interact for 10 min. All interactions were recorded with a digital video camera (iVIS HF S11/S21, Canon, Tokyo, Japan; Everio GZ-G5, Jvckenwood Corporation, Kanagawa Japan; or HC-W870M, Panasonic Corporation, Osaka, Japan). The following parameters were calculated from video recordings: percentage of females that spawned within the test period; number of courtship displays prior to spawning; number of wrapping attempts refused by the female; latency from the beginning of interaction to the first approach and courtship display by the male; latency from the first approach to the first courtship display by the male; latency from the beginning of interaction, first approach, and first courtship display to quivering.

**Analysis of neuronal size**. The cell body and nuclear sizes of FeSP neurons were measured as described previously[8]. In brief, whole female brains were fixed in 4% PFA, embedded in 5% agarose supplemented with 20% sucrose, and cut into 20 μm frozen sections in the coronal plane. Sections were incubated with a rabbit anti-Npb polyclonal antibody (RRID: AB_2810229), which has been shown to recognize Npba with high specificity[13], and then incubated with Alexa Fluor 488-conjugated goat anti-rabbit IgG (RRID: AB_2534114; Thermo Fisher Scientific). Neuronal cell bodies and nuclei were stained with NeuroTrace Fluorescent Nissl Stain (Neuro-Trace 530/615; Thermo Fisher Scientific) and DAPI (Roche Diagnostics), respectively. Fluorescent images were acquired with a confocal laser-scanning microscope (TCS SP8; Leica Microsystems), using excitation and emission wavelengths of 405 nm and 410–480 nm for DAPI, 488 nm and 495–545 nm for Alexa Fluor 488, and 552 nm and 620–700 nm for NeuroTrace 530/615.

The acquired images of the cell bodies and nuclei of Npb-immunoreactive neurons were converted to black and white binary images by thresholding using Adobe Photoshop (ver. 22; Adobe, San Jose, CA). The total areas of each cell body and nucleus were calculated by using ImageJ (https://imagej.nih.gov/ij/) and compared between genotypes.

**Patch-clamp recording**. For patch-clamp recordings, *ptger4b* knockout lines were crossed with the *npba*-GFP transgenic line[13]. Neuronal activities were recorded from GFP-labeled FeSP neurons in the PMm of *ptger4b*$^{+/+}$ and *ptger4b*$^{-/-}$ females, as previously reported with some minor modifications[12,52,53]. Briefly, whole brains removed from these females were hemisected along the midline and immersed in artificial cerebrospinal fluid (ACSF) comprising 134 mM NaCl, 2.9 mM KCl, 2.1 mM CaCl$_2$, 1.2 mM MgCl$_2$, 10 mM 4-(2-hydroxyethyl)-1-piper-azineethanesulfonic acid (HEPES), and 15 mM glucose (pH7.4, adjusted with NaOH). FeSP neurons were approached with patch pipettes made from borosilicate glass capillaries of 1.5-mm outer diameter (GD-1.5; Narishige, Tokyo, Japan) and, when filled with ACSF, had a resistance of 5–25 MΩ under an epifluorescent microscope (Eclipse FN1; Nikon, Tokyo, Japan). Targeted loose-patch recordings were performed with an Axopatch 200B amplifier (Molecular Devices, San Jose, CA) and digitized at 10 kHz using a Digidata 1550A and pClamp software (ver. 10.6; Molecular Devices). The hemi-brain preparations were continuously perfused with ACSF throughout experimentation. Action currents were recorded from FeSP neurons in voltage-clamp mode. Glutamate was applied focally to the recorded neuron by pressure ejection (500 kPa, 500 ms) from a glass pipette filled with ACSF containing 1 mM glutamate (L-glutamic acid monosodium salt hydrate; Sigma-Aldrich, St. Louis, MO) placed 10–30 μm from the cell body. The application of glutamate was made once or twice in a single recording, with an interval of at least 5 min between applications. ACSF without glutamate was applied to the control group. For experiments examining the effects of $PGE_2$ and EP4 antagonist, 1 μM of $PGE_2$ or GW 627368X in ACSF was applied by perfusion; ACSF containing only vehicle (ethanol) was used for control.

Data were analyzed using Clampfit software (ver. 10.7; Molecular Devices). A high-pass filter was applied to remove low-frequency background noise. Only those neurons that showed currents greater than 10 pA for a minimum of 10 min were included in the analysis. Firing frequency was calculated from 3 to 8 min segments of the recordings. For the analysis of the response to glutamate application, the

change in firing rate is expressed as the ratio of the average firing rate calculated from the 10 s period immediately after the glutamate puff application over the 10 s period immediately before.

**Immunohistochemistry.** Whole brains of $ptger4b^{+/+}$ and $ptger4b^{-/-}$ females were fixed in 4% PFA, embedded in paraffin, and cut into 10 μm coronal sections. After blocking with phosphate-buffered saline (PBS) containing 2% normal goat serum, sections were incubated overnight at 4 ℃ with the anti-Npba antibody described above[13] diluted 1:1000 in PBS containing 2% normal goat serum, 0.1% bovine serum albumin, and 0.02% keyhole limpet hemocyanin. The sections were then incubated overnight at 4 ℃ with Alexa Fluor 555-conjugated goat anti-rabbit IgG (RRID: AB_2535849; Thermo Fisher Scientific) and DAPI diluted 1:1000 in PBS. Images were captured as described above, using excitation and emission wavelengths of 405 nm and 410–480 nm for DAPI and 552 nm and 562–700 nm for Alexa Fluor 555. The immunofluorescence intensity of Npba was measured using LAS X software (ver. 3.7.4; Leica Microsystems).

**Statistics and reproducibility.** For continuous data, results are presented as mean ± standard error of the mean (SEM), with individual data points shown as dots except for neuronal size analysis, patch-clamp recordings, and Npba immunohistochemistry, where data are plotted as box-and-whisker plots by the Tukey method for visual clarity. Categorical data are presented as percentages.

Statistical analyses were performed using GraphPad Prism (ver. 8; GraphPad Software, San Diego, CA). Continuous data were compared between two groups by the unpaired two-tailed Student's $t$-test. Welch's correction was applied if the F-test indicated a significant difference in variance between groups. Differences in continuous data between more than two groups were evaluated by one-way analysis of variance (ANOVA) followed by either Bonferroni's (for comparisons among experimental groups) or Dunnett's (for comparisons of experimental versus control groups) post hoc test. Homogeneity of variance was verified for all data sets by Brown–Forsythe test. Two-way ANOVA followed by Bonferroni's post hoc test was used for analyses of diurnal fluctuations in brain $ptger4b$ expression, brain levels of $PGE_2$ and its metabolites, and the effect of EP4 antagonist on ovulation. Behavioral time-series data were analyzed using Kaplan–Meier plots with the inclusion of fish that did not exhibit the given behavior within the test period, following Jahn-Eimermacher et al.[54]. Differences between Kaplan–Meier curves were tested for statistical significance using Gehan–Breslow–Wilcoxon test with Bonferroni's correction. Statistical outliers in the behavioral data were determined with a ROUT test, using a false-positive rate ($Q$) of 0.1% and removed from the data sets. Fisher's exact test was used to compare categorical data.

**Reporting summary.** Further information on research design is available in the Nature Portfolio Reporting Summary linked to this article.

## Data availability
The RNA-seq data have been deposited in the DDBJ Sequenced Read Archive under the accession numbers DRR414603 to DRR414611. Source data for all graphs are provided in Supplementary Data 2. All other data supporting the findings of this study are available within the article and its supplementary information or can be provided from the corresponding author upon reasonable request.

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

## Acknowledgements

We thank National BioResource Project (NBRP) Medaka for providing the fosmid clone used in this study. We also thank Dr. Hideki Abe for technical advice on isolation and purification of FeSP neurons; Dr. Junpei Yamashita for performing ovariectomy; Dr. Yoshitaka Oka for useful discussions and providing the equipment for patch-clamp recordings; Yuji Nishiike and Kaito Ishikawa for overall experimental support; and Akira Hirata, Tomiko Iba, Kaoru Furukawa, and Ayu Kuwakubo for assistance with medaka husbandry. This work was supported by the Ministry of Education, Culture, Sports, Science, and Technology (MEXT) in Japan and the Japan Society for the Promotion of Science (JSPS) (MEXT/JSPS grant numbers 20J13802 (to TF), 16J03203 (to YK), 16H04979, 17H06429, and 19H03044 (to KO)).

## Author contributions

T.F., Y.K., and K.O. designed the research. T.F., Y.K., M.N., M.T., T.I., Y.O., T.B., and K.O. performed the research. Y.K., M.N., T.I., S.T., Y.S., D.S., M.S., Y.O., T.B., and K.M. contributed analytical tools. T.F., Y.K., M.N., M.T., T.I., S.T., Y.S., D.S., M.S., Y.O., T.B., K.M., and K.O. analyzed data. T.F., Y.K., and K.O. wrote the paper.

## Competing interests

The authors declare no competing interests.
