## [Peer Review File · Communications Biology]

Reviewers' comments:

Reviewer #1 (Remarks to the Author):

Ovarian-derived estrogens and prostaglandins affect female sexual behavior by acting through specific receptors in the female brain nervous system. However, the answers to questions such as which neurons are targeted by these ovarian hormones, and which receptors regulate what neural functions are still enigmatic. This paper by Okubo et al. examines these important questions. In the medaka, as in other species, it is the female that decides whether or not to mate. The mechanisms that control this behavior have important implications for selection and evolution, and this paper makes an important and exciting contribution.

The authors conducted a comprehensive analysis of gene expression changes in npba-positive cells (female-specific neurons) in the brain following ovarian removal, and found a group of peptide hormone genes that showed a significant decrease in their expression level. The authors focused on the female-specific npba-expressing neuronal population (FeSP) in the PMm/PMg region and generated a KO line for the prostaglandin receptor *ptger4b* expressed in this population. Based on the circadian cycle of medaka female reproduction, the authors carried out qPCR and LC-MS/MS analysis to examine the daily oscillation of *ptger4b* expression or production of PGE2 metabolites in a whole brain. These analyses detected diurnal variations in amount of these molecules. *Ptger4b* mutations did not affect body weight or ovarian development of female medaka, but did enhance their sexual receptivity. Electrophysiological analysis of FeSP neurons showed that the *ptger4b* mutants reduced spontaneous firing rate in a female-specific manner. Furthermore, npba immunohistochemistry of the FeSP neuronal population of this mutant suggested that the secretion of this peptide hormone was suppressed.

Major concerns

1. Regarding the qPCR and LC-MS/MS analysis to investigate the diurnal dynamics of *ptger4b* and PGE2:

Ptger4b is expressed in a cell type-specific manner in the brain. Based on this fact, qPCR analysis or metabolomic analysis of the entire brain reduces the sensitivity of detecting the expression level of the gene of interest and the amount of substances present.

If the goal is to investigate the reproductive cycle-dependent *ptger4b* gene expression and PGE2 metabolite abundance in females, the analysis should be done by cell-type or brain-region specific manner. Specifically, the authors should perform qPCR and LC-MS/MS analysis on three different cell groups: (1) Vv and PPa, which are common in males and females; (2) Pbl, which is male-specific; and (3) Ppp and PMm/PMg, which are female-specific. At the very least, a PMm/PMg-specific analysis, which is the focus of this study, should be performed (this can be done using npba-GFP).

It is also necessary to indicate at what time of day the brain samples used for the in situ hybridization analysis were harvested. In addition, if there is a time period when *ptger4b* expression in males exceeds that in females, in situ hybridization analysis should be performed on brains from that time period to confirm sexual dimorphism of the gene expression.

2. *Ptger4b*-deficient female phenotypes on ovarian function

pp.7, L189. "contrary to expectations based on previous findings"; It is difficult to say whether the results presented by the authors alone contradict previous findings. It is possible that other prostaglandin receptor may compensate for the ovarian functions. To clarify this possibility, pharmacological analysis of ovarian development and function using EP4 antagonist should be performed on the *ptger4b*-deficient females. This pharmacological experiment is also important to show the complementary function of other prostaglandin receptors in ovarian function.

3. Behavioral parameters for *ptger4b*-deficient female

Fig.5. h to j; Presented data showing the latency of behavioral expression are insufficient to determine in which behavioral parameters *ptger4b* plays an important role in female mating behavior. The duration of each behavioral parameter, "courtship displays by male," "wrappings by male," and

"quivering," should be quantified and compared among genotypes. This helps to identify the behavioral elements that are affected by the *ptger4b*-defect.

4. Spontaneous firing activity of FeSP neurons in *ptger4b*-deficient females

To investigate the spontaneous firing rate for a specific neuron such as FeSP neuron, the method employed by the authors is inadequate. To examine the spontaneous firing rate of a FeSP neuron, it needs to eliminate the influence of neural inputs from other neurons that connect to FeSP neurons. To eliminate this possibility, patch-clamp analysis of FeSP neurons should be performed under conditions where synaptic transmission is suppressed. Tetrodotoxin can be used for analysis under such conditions.

In order to describe the effect of *ptger4b* deficiency on the activity characteristics of FeSP neurons, in addition to the glutamate response analysis, the response of the FeSP neurons to the EP4 agonists or antagonists should be investigated. This experiment helps to clarify the role of *ptger4b* as a prostaglandin receptor in FeSP neurons.

Fig.6 e and f; In addition to the number of neurons tested, the number of animals should be indicated. If the data is obtained from only one individual animal, it is not a proper experiment. Data should be obtained from multiple animals.

Supplementary Fig.7 b; The spontaneous firing rate of FeSP neurons in *ptger4b*^{-/-} females before glutamate stimulation is comparable to that of wild type (+/+). This result is different from Fig.6D. The authors should explain why such a difference is occurring.

Minor

p.4, L.108; Appropriate references are needed for "PGE2 has been implicated in the regulation of female sexual receptivity across vertebrate phyla".

Fig.5 a; A typo "wappings" should be "wrappings".

Reviewer #2 (Remarks to the Author):

The work builds on previous findings that medaka female receptivity is linked to estrogen sensitive neurons (FeSP) in the preoptic area that express Neuropeptide b (Npba). In the present study, FeSP neurons from intact and ovariectomized female medaka were sequenced and found to express additional peptides as well as prostaglandin E2 receptor (*ptger4b*). Since prostaglandins are associated with female receptivity to males, the authors investigated whether these neurons could mediate the prostaglandin response. They developed a *ptger4b* CRISPR/CAS9 mutant which was normal in all reproductive respects investigated, except that females appeared to be more receptive towards males (shorter time to elicit quivering, a behaviour associated with spawning) and the FeSP firing intensity was reduced. Based on immunocytochemistry and in situ hybridization studies the authors conclude that the lower firing intensity results in lower Npba release. The work and conclusions are original and of general interest. However, although the work appears to be in general technically sound, some aspects including statistics and the conclusions would need to be better supported.

1- Does prostaglandin E2 increase or decrease receptivity in medaka? Since PGE2 has been shown to be involved in ovulation, one would expect that the increase in PGE2 associated with ovulation would increase receptivity. In goldfish, increased PGF2a levels is also related to ovulation and receptivity. However, although this was the initial assumption the results seem to conclude the opposite. It is curious that these estrogen dependent neurons would be involved in female receptivity considering that levels of estrogen at ovulation are usually low. I suggest that authors should try to demonstrate

the effect of PGE2 on females (intact/ovarectomized). The possible mechanism is not clear.

2- In addition, if PGE2 and metabolites in the brain are linked to ovulation and receptivity, why would males have higher levels? It would have been interesting to link to levels in blood plasma. I am not sure if one can say values of PGE2 fluctuate greatly if one considers the scale of the graphs. In females PGE2 does not vary more than about 1 -1.5 pg/mg, with n=3 and done only once it can be hardly considered "diurnal fluctuation" in Fig 3.

3- The link between neurone firing activity and Npba release although possible relies on quantification by immunocytochemistry and in situ hybridization, two methods that are not quite quantitative and designed to prove that reduction in firing activity led to accumulation of peptide. In Fig 6 E and F the overlap of error bars suggests the statistics is a bit surprising. If two standard error bars overlap for equal or nearly equal sample sizes the difference is not statistically significant ($P > 0.05$). This would mean that neither firing activity decreased nor immunofluorescence intensity increased.

4- Fig 2F and 2G- the main difference in the ESRs luciferase assay is variability. ESR2b data has little variability while ESR2a has high variability (technical issue?). Numbers of independent assays are not indicated but probably are low suggesting low statistical power and results questionable.

Reviewer #3 (Remarks to the Author):

This manuscript describes an interesting suite of experiments that detail the actions and roles of female-specific npba- expressing neurons in the forebrain of a species of a ricefish (Family Adrianchthyidae), the medaka, and specifically how this activity relates to female sexual receptivity. An impressive array of 5 experiments were conducted. First, the authors show that these neurons (which they have previously described on other publications) express a variety of neuropeptides and how this is altered by ovariectomy. Second, they examine the expression of the putative PGE2 receptor gene and show that it is also influenced by ovariectomy and sex steroid treatment, although not in an absolute (binary) manner suggesting other factors are at work. Third, they show that this receptor was expressed in a diurnal fashion, similar the reproductive cycle of this species (as well as many other physiological aspects). Fourth, they show that the expression of the PGE receptor did not appear to be influenced by normal ovarian function (a control for latter experiments). Fifth, they show that ptger4b-deficiency females exhibited increased sexual receptivity (latency to interact, quivering behaviour). Lastly, they show that ptger4b deficiency reduces spontaneous firing and neuropeptide release. Together, this suite of results strongly argue that ovarian estrogens influence FeSP neurons and expression of PGE2 receptors and that these changes in turn have effects(s) on the behaviour of female medaka including sexual receptivity. The work is very nicely and carefully performed although some simple experiments to measure PGE2 or inject it would have been welcome as well as studies to include other behaviours to clearly establish that the observed modulatory effects are exclusively sexual in nature.

This is excellent science. However, the question remains what is its real relevance beyond the medaka? Frankly, it is not clear to me. I do agree that this work shows that certain neural (genes) machinery has been conserved across evolution but I am confused to what extent and what exactly it means. There is much greater variation in vertebrates responsiveness to prostaglandins (PGs) than the authors acknowledge (see Guillette et al. 2019). This study does not in my opinion reveal "fundamental truths" about how female sexual receptivity is regulated all vertebrates, but rather describes some possibilities some of which might be examples of parallel evolution. I will give few examples. The authors mention "rigid control of sexual receptivity by ovarian excretion of estrogen, progestins and prostaglandins." Yet, many (thousands) of female vertebrates do not adopt sexually receptive behaviors unless males are actually present (suggesting visual, odor cues, etc. have major role - it is not just endocrine or "rigid"). Further, studies of the goldfish and other cypriniform fishes show this clearly and also show that PGF2a has a key role that it is NOT associated with the ovary but rather the ovarian tract (injected egg substitutes trigger female receptivity and hyophysectomy does

not influence responses – see Stacey & Liley 1974 (Nature) and Sorensen et al. 2018 (Gen. Comp Endocrinol). The cypriniformes are a group of nearly 5000 species compared to the mere 36 species found in the ricefish family to which the medaka belongs. It is hard to see it is any especially great evolutionary model (like any fish for that matter- the fishes are very diverse group!). Further, all studies of prostaglandin (PG) action in cypriniformes show that PGF2a has strong, immediate and highly specific actions on female receptivity so this seems common in this group (Stacey & Goetz 1982; Munakata & Kobayashi 2010). Notably, PGE2 seems to have no activity (Stacey & Peter 1979) and cypriniform females also are sexually inactive until PGF2a levels rise and male present and courting them (Sorensen et al. 2018). So what might the role of a strong and universal inhibitory factor such as PGE actually be and why is it needed by female fish (how might evolution favor it)? Similar (but not identical) actions of PGFs are seen in the family cichlidae (about 2000 species) (Juntii et al. 2016) but here estradiol seems to be required as well as it is the gouramis. Variation in PG function is expected because these fishes are often nest guards and they have had well over 100 million year to evolve. The argument generally is that PGFs are produced by the ovarian tract associated with ovulation and the fact that eggs in most fishes are not viable for long after they have ovulated. PGE2 is the precursor for PGF2a, yet the possible/likely relationships between these lipids is never addressed in this manuscript either as circulatory hormones or neurotransmitters, or perhaps both? Although poorly understood, a variety of responses to injected PGs have been seen in other female fishes (the vast majority of the vertebrates). The authors do state that “Of note, these prostaglandins also have facilitative effects on ... many other species, but inhibitory on others” but the authors never really account for how this could possibly be and how PGE2 could reflect a singular conserved neuroendocrine mechanism as they suggest. Mammals diverged from fishes some 400 million years ago, show very different behaviors and physiologies, likely related to processes such as live-bearing, egg viability, internal vs external fertilization, maternal behaviors, etc.

In conclusion, while I find the present work very interesting and compelling and do commend the authors for their meticulous work, I am simply not convinced what exactly it means to neuroendocrine control of female vertebrate behavior. I do hope to see this work published but the context needs to be reconsidered and adjusted in my opinion.

Responses to the Reviewers

First of all, we would like to thank the three anonymous reviewers for the positive evaluation of our work and the constructive comments and suggestions, which significantly helped improve the manuscript. We greatly appreciate their time and effort. We would also like to thank Professor Ross Bathgate, an Editorial Board Member of *Communications Biology*, for allowing us to submit a revised manuscript. We have considered the reviewers' comments and suggestions carefully and made revisions accordingly. In the revised manuscript, revised portions are indicated in **red letters**. In this document, the original text is indicated in **blue letters** and the revised text is in **red letters**. All line numbers in this document refer to those in the revised manuscript.

Reviewer #1

Reviewer #1's overall comment:

Ovarian-derived estrogens and prostaglandins affect female sexual behavior by acting through specific receptors in the female brain nervous system. However, the answers to questions such as which neurons are targeted by these ovarian hormones, and which receptors regulate what neural functions are still enigmatic. This paper by Okubo et al. examines these important questions. In the medaka, as in other species, it is the female that decides whether or not to mate. The mechanisms that control this behavior have important implications for selection and evolution, and this paper makes an important and exciting contribution.

The authors conducted a comprehensive analysis of gene expression changes in npba-positive cells (female-specific neurons) in the brain following ovarian removal, and found a group of peptide hormone genes that showed a significant decrease in their expression level. The authors focused on the female-specific npba-expressing neuronal population (FeSP) in the PMm/PMg region and generated a KO line for the prostaglandin receptor *ptger4b* expressed in this population. Based on the circadian cycle of medaka female reproduction, the authors carried out qPCR and LC-MS/MS analysis to examine the daily oscillation of *ptger4b* expression or production of PGE2 metabolites in a whole brain. These analyses detected diurnal variations in amount of these molecules. *Ptger4b* mutations did not affect body weight or ovarian development of female medaka, but did enhance their sexual receptivity. Electrophysiological analysis of FeSP neurons showed that the *ptger4b* mutants reduced spontaneous firing rate in a female-specific manner. Furthermore, npba immunohistochemistry of the FeSP neuronal population of this mutant suggested that the secretion of this peptide hormone was suppressed.

Response to reviewer #1's overall comment:

We thank the reviewer for this very positive evaluation of our work and greatly appreciate their helpful comments for improving the manuscript.

Reviewer #1's specific comment 1:

1. Regarding the qPCR and LC-MS/MS analysis to investigate the diurnal dynamics of *ptger4b* and PGE2:

Ptger4b is expressed in a cell type-specific manner in the brain. Based on this fact, qPCR analysis or metabolomic analysis of the entire brain reduces the sensitivity of detecting the expression level of the gene of interest and the amount of substances present.

If the goal is to investigate the reproductive cycle-dependent *ptger4b* gene expression and PGE₂ metabolite abundance in females, the analysis should be done by cell-type or brain-region specific manner. Specifically, the authors should perform qPCR and LC-MS/MS analysis on three different cell groups: (1) Vv and PPa, which are common in males and females; (2) Pbl, which is male-specific; and (3) Ppp and PMm/PMg, which are female-specific. At the very least, a PMm/PMg-specific analysis, which is the focus of this study, should be performed (this can be done using npba-GFP).

It is also necessary to indicate at what time of day the brain samples used for the *in situ* hybridization analysis were harvested. In addition, if there is a time period when *ptger4b* expression in males exceeds that in females, *in situ* hybridization analysis should be performed on brains from that time period to confirm sexual dimorphism of the gene expression.

Response to reviewer #1's specific comment 1:

We agree with the notion that changes in *ptger4b* expression/PGE₂ levels within specific brain nuclei are more relevant to understanding the behavioral role of PGE₂/Ptger4b than the overall levels in the whole brain. Unfortunately, however, it is not possible to measure PGE₂ levels within individual nuclei of the medaka brain, as obtaining the necessary amount of tissue for the measurement is simply not feasible. Whole brains from 3 medaka need to be pooled to obtain sufficient tissue to measure PGE₂ levels by LC-MS/MS analysis. The current data set was obtained by measuring 5 samples per sex at 6 time points, requiring 180 fish in total. Performing the experiment proposed by this reviewer would require thousands of medaka, even with a smaller sample size. Moreover, the brain nuclei mentioned by this reviewer are very small and lie in proximity to one another, rendering the dissection of these nuclei unreliable.

As an alternative approach, we attempted to determine the brain nuclei that actively synthesize prostaglandins during the spawning period by analyzing the expression of *ptgs2*, the gene encoding cyclooxygenase 2 (COX2), by *in situ* hybridization. However, the expression level of this gene in the medaka brain was found to be below the detection limit of *in situ* hybridization. This is in accordance with another study which found relatively low expression levels of COX enzymes in the medaka brain (Fujimori *et al.*, 2011, *Mol Cell Endocrinol*, 332:67–77). We would like to note that this does not necessarily mean that PGE₂ synthesis in the medaka brain is insignificant or unimportant. It merely means that PGE₂ synthesis would occur with relatively low levels of COX enzymes.

We strongly agree with this reviewer that it is important to quantify *ptger4b* expression levels in different brain nuclei by sex and time. As described above, dissecting each individual brain nuclei and performing real-time PCR is not possible, which leaves *in situ* hybridization as our only option. However, as can be seen in the micrographs in Figure 3c, the hybridization signal for *ptger4b* was very faint and difficult to quantify by the conventional method. In order to respond to this comment, therefore, we have tried several methods to amplify the hybridization signal to quantifiable levels, including the TSA Plus signal amplification technology (TSA Plus Fluorescein System; PerkinElmer Marlborough, MA), Tyramide SuperBoost technology (Alexa Fluor 555 Tyramide SuperBoost Kit; Thermo Fisher Scientific, Waltham, MA), and signal amplification using a secondary antibody (RRIDs: AB_10680417, Abcam, Cambridge, UK). Unfortunately, all of these methods resulted in high backgrounds and did not improve the signal-to-noise ratio. Currently, the methods available to us do not allow us to quantify the expression of *ptger4b* in each brain nucleus by sex and time. We greatly apologize for this and hope that this reviewer understands that we have done our best to address their comment.

Line 376: Brains sampled at 1 to 3 hours after light onset (halo) were used for *in situ* hybridization analysis, which is noted in the Animals subsection of Methods as “Fish were sampled at 1 to 3 halo in all analyses”. Note that, as shown in Figure 3d, the male brain showed higher overall expression of *ptger4b* than the female brain at this time period, and indeed, male-specific *ptger4b* expression was observed in the Pbl, as shown in Figure 3c. This actually confirms the sexual dimorphism in *ptger4b* expression as suggested by this reviewer.

Reviewer #1's specific comment 2:

2. Ptger4b-deficient female phenotypes on ovarian function

pp.7, L189. “contrary to expectations based on previous findings”; It is difficult to say whether the results presented by the authors alone contradict previous findings. It is possible that other prostaglandin receptor may compensate for the ovarian functions. To clarify this possibility, pharmacological analysis of ovarian development and function using EP4 antagonist should be performed on the *ptger4b*-deficient females. This pharmacological experiment is also important to show the complementary function of other prostaglandin receptors in ovarian function.

Response to reviewer #1's specific comment 2:

We appreciate this comment and agree that our results do not exclude the possibility that other EP4 subtypes may compensate for the loss of Ptger4b and help to induce ovulation. To examine this possibility, we have investigated the effect of an EP4 antagonist (GW 627368X) on ovulation using an *in vitro* assay developed by Fujimori *et al.* (2011, *Mol Cell Endocrinol*, 332:67–77; 2022, *Mol Cell Endocrinol*, 362:76–84). The results showed that the EP4 antagonist did not inhibit ovulation, not only in *ptger4b*-deficient females, but also in wild-type and heterozygous females (please see the figure below), in contrast to what was reported by Fujimori *et al.* (2011; 2012). Although the reason for this discrepancy remains unclear (it may be due to different strains of medaka used), our results, combined with the fact that *ptger4b*-deficient females ovulate normally, suggest that Ptger4b is not essential for ovulation in medaka.

To include the results of this additional analysis, we have revised the manuscript as follows.

1) Line 189: The following line has been added to the Results section. “We further investigated the effect of the EP4 antagonist GW 627368X on ovulation using an *in vitro* assay as previously reported^{16, 17} in order to examine the possibility that other EP4 subtypes may compensate for the loss of Ptger4b and help to induce ovulation. The results showed that GW 627368X failed to inhibit ovulation not only

in *ptger4b*^{-/-} females, but also in *ptger4b*^{+/+} and *ptger4b*^{+/-} females, in contrast to previous reports^{16, 17} (Fig. 4h).”.

2) The following reference, which is cited in the above text, has been added to the reference list (reference No. 17).

“Fujimori, C., Ogiwara, K., Hagiwara, A., & Takahashi, T. New evidence for the involvement of prostaglandin receptor EP4b in ovulation of the medaka, *Oryzias latipes*. *Mol. Cell. Endocrinol.* **332**, 76–84 (2012).”

As a consequence, references 18–50 have been renumbered to 19–51 in the text and in the reference list.

3) Line 560: The following subsection has been added to the Methods section.

“*In vitro* ovulation assay

An *in vitro* ovulation assay was performed according to the method of Fujimori et al^{16, 17}. In brief, ovaries were removed from *ptger4b*^{+/+}, *ptger4b*^{+/-}, and *ptger4b*^{-/-} females of the $\Delta 17$ knockout line at 12 halo. All large-sized ovarian follicles were isolated from each ovary, split evenly into control and treatment groups, and incubated in 90% Medium 199 with Earle’s salts (Thermo Fisher Scientific) containing 50 mg/ml gentamycin and either vehicle (dimethyl sulfoxide) alone or 20 μ M of the EP4 antagonist GW 627368X (Cayman Chemical Company, Ann Arbor, MI), respectively. The number of oocytes that successfully ovulated was counted the following day at 5 halo. The ovulation rate was presented as the percentage of oocytes that ovulated.”.

4) Line 650: The sentence for statistical analysis “Two-way ANOVA followed by Bonferroni’s *post hoc* test was used for analyses of diurnal fluctuations in brain *ptger4b* expression and brain levels of PGE₂ and its metabolites.” has been changed to “Two-way ANOVA followed by Bonferroni’s *post hoc* test was used for analyses of diurnal fluctuations in brain *ptger4b* expression, brain levels of PGE₂ and its metabolites, and the effect of EP4 antagonist on ovulation.”

5) Fig. 4: A new panel showing the above graph (panel h) has been added and the layout of the other panels has been modified accordingly. The revised Fig. 4 is shown below.

6) Line 890: The following line has been added to the legend for Fig. 4. “**h** Effect of the EP4 antagonist GW 627368X on the ovulation of oocytes from *ptger4b*^{+/+}, *ptger4b*^{+/-}, and *ptger4b*^{-/-} females *in vitro* (n = 5 for each experimental group except *ptger4b*^{+/-} females treated with the EP4 antagonist, where n = 6; each n comprises 8–20 oocytes).”

7) Line 893: “Bonferroni’s *post hoc* test (c, d, f, g)” has been changed to “Bonferroni’s *post hoc* test (c, d, f–h)”.

Reviewer #1’s specific comment 3:

3. Behavioral parameters for *ptger4b*-deficient female

Fig.5. h to j; Presented data showing the latency of behavioral expression are insufficient to determine in which behavioral parameters *ptger4b* plays an important role in female mating behavior. The duration of each behavioral parameter, "courtship displays by male," "wrappings by male," and "quivering," should be quantified and compared among genotypes. This helps to identify the behavioral elements that are affected by the *ptger4b*-defect.

Response to reviewer #1’s specific comment 3:

We believe there may be a misconception about medaka mating behavior or the meaning on the data in Fig. 5. The courtship displays and wrappings by males are very quick behaviors, often completed within a second or less. Here is an excellent YouTube video which clearly shows the steps of medaka mating behavior (<https://www.youtube.com/watch?v=7HB5UaNrTSQ>). In Fig. 5h–j we show a significant reduction in latency to quivering from the beginning of the interaction (h), the first approach by the male (i), and from the first courtship display by the male (j) in *ptger4b*-deficient females. The reduction in all of these parameters strongly suggests that these females require less time for the process of perceiving and evaluating male courtship, or the subsequent motor response.

Line 219: This misconception was probably caused by our inadequate explanation, and the text in the Results section “*ptger4b* deficiency has a female-specific effect of increasing sexual receptivity towards male courtship.” has been edited to read “*ptger4b* deficiency has a female-specific effect of increasing sexual receptivity towards male courtship, most likely by shortening the process of perceiving and evaluating male courtship or the subsequent motor response.” accordingly.

Reviewer #1’s specific comment 4:

4. Spontaneous firing activity of FeSP neurons in *ptger4b*-deficient females

To investigate the spontaneous firing rate for a specific neuron such as FeSP neuron, the method employed by the authors is inadequate. To examine the spontaneous firing rate of a FeSP neuron, it needs to eliminate the influence of neural inputs from other neurons that connect to FeSP neurons. To eliminate this possibility, patch-clamp analysis of FeSP neurons should be performed under conditions where synaptic transmission is suppressed. Tetrodotoxin can be used for analysis under such conditions. In order to describe the effect of *ptger4b* deficiency on the activity characteristics of FeSP neurons, in addition to the glutamate response analysis, the response of the FeSP neurons to the EP4 agonists or antagonists should be investigated. This experiment helps to clarify the role of *ptger4b* as a prostaglandin receptor in FeSP neurons.

Fig.6 e and f; In addition to the number of neurons tested, the number of animals should be indicated. If the data is obtained from only one individual animal, it is not a proper experiment. Data should be obtained from multiple animals.

Supplementary Fig.7 b; The spontaneous firing rate of FeSP neurons in *ptger4b*^{-/-} females before glutamate stimulation is comparable to that of wild type (+/+). This result is different from Fig.6D. The authors should explain why such a difference is occurring.

Response to reviewer #1’s specific comment 4:

As this reviewer suggests, it may be useful to investigate the cell-autonomous firing of FeSP neurons under conditions where synaptic inputs from other neurons are eliminated. However, tetrodotoxin (TTX), which is used to eliminate synaptic inputs, blocks TTX-sensitive voltage-gated sodium channels, thus inhibiting the firing of action potentials in nearly all neurons. Therefore, spontaneous firing of FeSP neurons cannot be measured in the presence of TTX. What can be measured with TTX application is miniature excitatory/inhibitory postsynaptic potentials, which are caused by the neurotransmitter-mediated opening/closing of ion channels at postsynaptic sites. This is a neural property that differs significantly from spontaneous firing activity and is beyond the scope of the present study.

We agree with the second point raised by this reviewer and in response, we additionally examined the effects of bath application of PGE₂ and an EP4 antagonist on the firing rate of FeSP neurons. The results showed that the firing rate of FeSP neurons was not acutely altered by either PGE₂ or an EP4 antagonist, as shown in the graph below.

Based on the fact that FeSP neurons do not show diurnal changes in firing rate (Kikuchi *et al.*, 2019, *Endocrinology*, 160:827–839) despite diurnal fluctuations in brain PGE₂ levels (Fig. 3e), we argued that the effects of PGE₂/EP4 signaling on FeSP neurons would be constitutive rather than acute. The new data above strongly supports this argument. We appreciate this discerning comment from reviewer #1.

To include these results and discussion, we have revised the manuscript as follows.

1) Line 252: The following line has been added to the Results section: “We additionally assessed the effects of PGE₂ and the EP4 antagonist GW 627368X on the firing activity of FeSP neurons. FeSP neurons showed no change in spontaneous firing rate in response to bath application of either PGE₂ or GW 627368X (Supplementary Fig. 7d). These results suggest that the effects of PGE₂/Ptger4b signaling on FeSP neurons are constitutive rather than acute.”

2) Line 316: The sentence in the Discussion section “Interestingly, given that the firing rates of FeSP neurons do not correspond to the diurnal fluctuation of brain PGE₂ levels, the effects of PGE₂/Ptger4b signaling on the electrophysiological properties of FeSP neurons are seemingly constitutive, rather than acute and transient, unlike those typically seen in mammalian peptidergic neurons^{26, 30}.” has been modified and now reads “Interestingly, given that the firing rates of FeSP neurons are not altered by acute treatment with PGE₂/EP4 antagonist and do not correspond to the diurnal fluctuation of brain PGE₂ levels, the effects of PGE₂/Ptger4b signaling on the electrophysiological properties of FeSP neurons are seemingly constitutive, rather than acute and transient. These effects are in contrast to those in mammalian peptidergic neurons, where PGE₂/EP4 signaling typically mediates acute changes in excitability^{27, 31}.”.

3) Line 617: The following line has been added to the Methods section: “For experiments examining the effects of PGE₂ and EP4 antagonist, 1 μM of PGE₂ or GW 627368X in ACSF was applied by perfusion; ACSF containing only vehicle (ethanol) was used for control.”

4) Supplementary Fig. 7: A new panel showing the above graph (panel d) has been added. The revised Supplementary Fig. 7 is shown below.

5) Lines 1002 and 1005: The following line has been added to the legend for Supplementary Fig. 7. **“d Ratio of the firing rates of FeSP neurons before and after the application of medium only (n = 10 from 8 individuals), PGE₂ (n = 8 from 8 individuals), or the EP4 antagonist GW 627368X (n = 4 from 3 individuals) in *ptger4b*^{+/+} females.”** and **“and Dunnett’s *post hoc* test (versus medium control) (d)”**

All morphological and electrophysiological data on FeSP neurons were obtained from at least 3 individuals, thus eliminating this reviewer’s concern. The number of animals used for each experiment has been added to the figure legends for Fig. 6, Supplementary Fig. 6, and Supplementary Fig. 7 as follows.

- 1) Line 911: “for each genotype” has been changed to “from 5 individuals for each genotype”.
- 2) Line 916: “from 10 individuals” has been added.
- 3) Line 916: “from 14 individuals” has been added.
- 4) Line 917: “from 3 individuals” has been added.
- 5) Line 918: “from 3 individuals” has been added.
- 6) Line 986: “for each genotype” has been changed to “from 5 individuals for each genotype”.
- 7) Line 990: “from 7 individuals” has been added.
- 8) Line 991: “from 7 individuals” has been added.
- 9) Line 998: “n=7” has been changed to “n = 7 neurons from 4 individuals”.
- 10) Line 998: “neurons from 8 individuals” has been added.
- 11) Line 1001: “neurons from 8 individuals” has been added.
- 12) Line 1001: “neurons from 14 individuals” has been added.

Supplementary Fig. 7: It was our oversight to present a trace from the *ptger4b*^{-/-} recording in Supplementary Fig. 7b that did not adequately reflect the data in Figure 6d. In response to this comment, the trace of the *ptger4b*^{-/-} recording has been changed to better reflect the data in Figure 6d. Please see the edited figure below.

Reviewer #1’s minor comment 1:

p.4, L.108; Appropriate references are needed for “PGE2 has been implicated in the regulation of female sexual receptivity across vertebrate phyla”.

Response to Reviewer #1’s minor comment 1:

Line 109: We have cited Guillette *et al.* (1991, *Am. J. Physiol.* 260:854–861) (reference No. 10) accordingly.

Reviewer #1’s minor comment 2:

Fig.5 a; A typo “wappings” should be “wrappings”.

Response to Reviewer #1’s minor comment 2:

Fig. 5: This error has been corrected. Please see the corrected figure below.

Reviewer #2

Reviewer #2’s overall comment:

The work builds on previous findings that medaka female receptivity is linked to estrogen sensitive neurons (FeSP) in the preoptic area that express Neuropeptide b (Npba). In the present study, FeSP neurons from intact and ovariectomized female medaka were sequenced and found to express additional peptides as well as prostaglandin E2 receptor (ptger4b). Since prostaglandins are associated with female receptivity to males, the authors investigated whether these neurones could mediate the prostaglandin response. They developed a ptger4b CRISPR/CAS9 mutant which was normal in all reproductive respects investigated, except that females appeared to be more receptive towards males (shorter time to elicit quivering, a behaviour associated with spawning) and the FeSP firing intensity was reduced. Based on immunocytochemistry and in situ hybridization studies the authors conclude that the lower firing intensity results in lower Npba release. The work and conclusions are original and of general interest. However, although the work appears to be in general technically sound, some aspects including statistics

and the conclusions would need to be better supported.

Response to reviewer #2's overall comment:

We thank this reviewer for their constructive comments, which we found very useful to improve the manuscript. Below is a point-by-point response to the comments.

Reviewer #2's specific comment 1:

1- Does prostaglandin E2 increase or decrease receptivity in medaka? Since PGE2 has been shown to be involved in ovulation, one would expect that the increase in PGE2 associated with ovulation would increase receptivity. In goldfish, increased PGF2a levels is also related to ovulation and receptivity. However, although this was the initial assumption the results seem to conclude the opposite. It is curious that these estrogen dependent neurons would be involved in female receptivity considering that levels of estrogen at ovulation are usually low. I suggest that authors should try to demonstrate the effect of PGE2 on females (intact/ovarectomized). The possible mechanism is not clear.

Response to reviewer #2's specific comment 1:

We recently found that female medaka deficient for an estrogen receptor, *Esr2b*, are not receptive to males, despite retaining normal ovarian function (Nishiike *et al.*, 2021, *Curr Biol*, 31:1699–1710). This clearly indicates that estrogen signaling is essential for female receptivity in medaka. If estrogen plays a priming role rather than being directly responsible for female receptivity in teleosts, as it does in mammals, then sexual receptivity does not necessarily coincide with elevated levels of circulating estrogen. Nonetheless, female medaka exhibit high levels of circulating estrogen at the onset of the light period, when they ovulate and spawn (Soyano *et al.*, 1993, *Fish Physiol Biochem*, 11:265–272; Kayo *et al.*, 2020, *Gen Comp Endocrinol*, 285:113272). Thus, whatever the mode of action of estrogen in teleosts, there is no contradiction in the view that estrogen-dependent neurons are involved in female receptivity in medaka.

As this reviewer suggests, it may be useful to determine the effect of PGE₂ on sexual receptivity of female medaka by administering it *in vivo*. Therefore, we preliminarily treated female medaka with PGE₂ (100 ng/ml; immersion in water for 24 hours) but found no noticeable changes in their sexual receptivity as shown in the graph below.

Some evidence in other teleost species suggests that in teleosts, peripherally administered PGE₂ is not efficiently transported into the brain. In goldfish, for example, it has been reported that injection of a high concentration of PGE₂ into the third ventricle stimulated spawning behavior, but intraperitoneal injection produced only a marginal effect (Stacey 1976, *Prostaglandins*, 12:113–126; Stacey and Peter,

1979, *Physiol Behav*, 22:1191–1196). In zebrafish, PGE₂ has been shown to have no effect on mating behavior when administered to the water (Pradhan and Olsson, 2015, *Behav Brain Funct*, 11:23). Therefore, in order to study the general effects of PGE₂ on mating behavior in teleosts, we would be required to inject PGE₂ directly into the brain. Unfortunately, because medaka are small in size and vulnerable to head surgery, we currently have no reliable methods of administering drugs directly into the brain without affecting their subsequent performance of mating behavior (we have tried several times, but all have failed). We would be very grateful if this reviewer could understand the practical difficulties of such an experiment, although we do agree it would be of general interest.

In addition, a previous study has shown that there is no diurnal variation in PGE₂ levels in the medaka ovary, and instead ovulation is regulated at the level of the receptor (Fujimori *et al.*, 2011, *Mol Cell Endocrinol*, 332:67–77). Because there is no increase in PGE₂ associated with ovulation, we would not expect it to drive subsequent mating behavior. This and the aforementioned studies in the previous paragraph suggest that the brain is the primary source of behaviorally relevant PGE₂, which is independent from ovarian PGE₂. Thus, it does not appear that the roles for PGE₂ in ovulation and mating behavior are incompatible.

Reviewer #2's specific comment 2:

2- In addition, if PGE₂ and metabolites in the brain are linked to ovulation and receptivity, why would males have higher levels? It would have been interesting to link to levels in blood plasma. I am not sure if one can say values of PGE₂ fluctuate greatly if one considers the scale of the graphs. In females PGE₂ does not vary more than about 1 -1.5 pg/mg, with n=3 and done only once it can be hardly considered "diurnal fluctuation" in Fig 3.

Response to reviewer #2's specific comment 2:

A previous study has shown that there is no diurnal variation in PGE₂ levels in the medaka ovary (Fujimori *et al.*, 2011, *Mol Cell Endocrinol*, 332:67–77). Instead, the expression of *ptger4b* is greatly upregulated in the ovary prior to ovulation, meaning regulation occurs at the level of the receptor (Fujimori *et al.*, 2011, *Mol Cell Endocrinol*, 332:67–77). We therefore do not believe that changes in brain PGE₂ levels reflect changes in ovarian production or serum concentration. Although it may be of interest to measure serum prostaglandin levels, we do not expect this experiment to yield major results and, moreover, due to the small size of medaka, collecting sufficient amounts of serum for this measurement would not be feasible. As this reviewer has mentioned, males have significantly higher brain PGE₂ levels at the beginning of the light period. This indicates that the male brain has a higher rate of prostaglandin synthesis than the female brain at this timepoint, but it is unclear what the functional significance of this would be. We agree that this is an interesting finding and that it could be a topic for future research.

There may be a misconception about the number of samples used to measure brain PGE₂ levels. We state “n = 5 per sex and time, except males at 4 hours after light onset, where n = 4” in the legend for Fig. 3 (line 873) and “the brains from 3 fish were pooled per sample” in the Methods section (line 523). 5 samples comprising 3 whole brains were measured per timepoint per sex, resulting in a total of 15 individuals from which data were obtained. In addition, we obtained similar results in a preliminary experiment with samples (albeit small in number) independent of those used for the present data, and we believe our results are reliable.

Line 172: PGE₂ levels in the female brain varied from 1.44 pg/mg (at 0 hours after light onset) to 3.29 pg/mg (at 20 hours after light onset), which is a larger variation (2.28-fold) than this reviewer noted. However, we agree with the reviewer that it is an overstatement to describe this degree of variation as “PGE₂ levels fluctuate greatly”. We therefore have changed this phrase to “PGE₂ levels show diurnal fluctuations”.

Reviewer #2’s specific comment 3:

3- The link between neurone firing activity and Npba release although possible relies on quantification by immunocytochemistry and in situ hybridization, two methods that are not quite quantitative and designed to prove that reduction in firing activity led to accumulation of peptide. In Fig 6 E and F the overlap of error bars suggests the statistics is a bit surprising. If two standard error bars overlap for equal or nearly equal sample sizes the difference is not statistically significant (P>0.05). This would mean that neither firing activity decreased nor immunofluorescence intensity increased.

Response to reviewer #2’s specific comment 3:

As of now, there are no available methods for the direct quantification neuropeptide release *in vivo* during the performance of behaviors in any species. Quantification of Npba immunofluorescence in FeSP neurons was therefore the only option available to us. We do agree, however, that *in vitro* characterization of neuropeptide release from FeSP neurons is of interest and could be a topic for future research.

The graphs in Fig. 6e and f are box-and-whisker plots based on the Tukey method, not bar graphs. What look like error bars in box-and-whisker plots are “whiskers”, which represent the distribution outside the central quartiles. For reference, we have presented below the data from Fig. 6e and f in bar graphs showing the mean \pm standard error of the mean. As can be seen, there is no overlap of the error bars. We hope this dispels any concerns this reviewer may have over the accuracy of our statistical analysis.

Line 640: We realized that we had not stated in the “Statistics and reproducibility” subsection of Methods that the Npba immunohistochemistry data were represented in a box-and-whisker plot, which may have caused this confusion. In response, “For continuous data, results are presented as mean \pm standard error of the mean (SEM), with individual data points shown as dots; for neuronal size analysis and patch-clamp recordings, data are plotted as box-and-whisker plots by the Tukey method for visual clarity.” has been edited to read “For continuous data, results are presented as mean \pm standard error of the mean (SEM), with individual data points shown as dots except for neuronal size analysis, patch-

clamp recordings, and Npba immunohistochemistry, where data are plotted as box-and-whisker plots by the Tukey method for visual clarity.”.

Reviewer #2's specific comment 4:

4- Fig 2F and 2G- the main difference in the ESRs luciferase assay is variability. ESR2b data has little variability while ESR2a has high variability (technical issue?). Numbers of independent assays are not indicated but probably are low suggesting low statistical power and results questionable.

Response to reviewer #2's specific comment 4:

We state that “Each assay was performed in triplicate and repeated independently three times.” in the Transcriptional activity assay subsection of Methods (line 498). However, as this reviewer has pointed out, this information is missing from the figure legend. To correct this, we have added this information to the legend of Fig. 2 (please see the next page for details).

We do agree that the data for ESR2a showed an unusually large degree of variability, especially when compared to ESR2b. In response, we have repeated the assay for ESR2a and doubled the number of samples. The new data set presented below shows much less variation than the previous set, and induction by E2 is now significant at concentrations above 10^{-9} M. We would like to note that this new set of data does not influence the overall interpretation of our results because FeSP neurons do not express ESR2a, only ESR1 and ESR2b.

We have accordingly updated the relevant sections of the manuscript as follows.

1) Line 127: The sentences in the Results section “E₂ induced a significant increase in luciferase activity in the presence of ESR2b ($p < 0.0001$ at all concentrations tested) (Fig. 2f). Although not significant, slight inhibition and induction were observed in the presence of ESR1 and ESR2a, respectively (Fig. 2f).” has been modified and now reads “E₂ induced a significant increase in luciferase activity in the presence of ESR2a ($p = 0.0056, 0.0024, 0.0015,$ and 0.0024 at $10^{-9}, 10^{-8}, 10^{-7},$ and 10^{-6} M, respectively) and ESR2b ($p < 0.0001$ at all concentrations tested) (Fig. 2f). Although not significant, a slight inhibition was observed in the presence of ESR1 (Fig. 2f).”.

2) Line 498: The sentence in the Methods section “Each assay was performed in triplicate and repeated independently three times.” has been modified and now reads “Each assay was performed in triplicate and repeated independently three times, except for the assay with ESR2a, which was repeated six times.”.

3) The graph in Fig. 2f has been replaced with the new data shown above.

4) Lines 857 and 860: We have added “Each assay was performed in triplicate and repeated independently three times, except for the dose-response assay with Esr2a, which was repeated six times.” and “** $p < 0.01$;” to the legend of Fig. 2.

Reviewer #3

Reviewer #3's comment 1:

This manuscript describes an interesting suite of experiments that detail the actions and roles of female-specific npba- expressing neurons in the forebrain of a species of a ricefish (Family Adrianchthyidae), the medaka, and specifically how this activity relates to female sexual receptivity. An impressive array of 5 experiments were conducted. First, the authors show that these neurons (which they have previously described on other publications) express a variety of neuropeptides and how this is altered by ovariectomy. Second, they examine the expression of the putative PGE2 receptor gene and show that it is also influenced by ovariectomy and sex steroid treatment, although not in an absolute (binary) manner suggesting other factors are at work. Third, they show that this receptor was expressed in a diurnal fashion, similar the reproductive cycle of this species (as well as many other physiological aspects). Fourth, they show that the expression of the PGE receptor did not appear to be influenced by normal ovarian function(a control for latter experiments). Fifth, they show that ptger4b-defincy females exhibited increased sexual receptivity (latency to interact, quivering behaviour). Lastly, they show that ptger4b deficiency reduces spontaneous firing and neuropeptide release. Together, this suite of results strongly argue that ovarian estrogens influence FeSP neurons and expression of PGE2 receptors and that these changes in turn have effects(s) on the behaviour of female medaka including sexual receptivity. The work is very nicely and carefully performed although some simple experiments to measure PGE2 or inject it would have been welcome as well as studies to include other behaviours to clearly establish that the observed modulatory effects are exclusively sexual in nature.

Response to reviewer #3's comment 1:

We thank the reviewer for their positive evaluation of our work and thoughtful comments, which we have used to improve the manuscript.

In response to this comment and reviewer #2's specific comment 1, we preliminarily treated female medaka with PGE₂ (100 ng/ml; immersion in water for 24 hours) but found no noticeable changes in their sexual receptivity as shown in the graph below.

Some evidence in other teleost species suggests that in teleosts, peripheral PGE₂ is not efficiently transported into the brain to affect mating behavior. In goldfish, for example, it has been reported that injection of a high concentration of PGE₂ into the third ventricle stimulated spawning behavior, but intraperitoneal injection produced only a marginal effect (Stacey 1976, *Prostaglandins*, 12:113–126; Stacey and Peter, 1979, *Physiol Behav*, 22:1191–1196). In zebrafish, PGE₂ has been shown to have no effect on mating behavior when administered to the water (Pradhan and Olsson, 2015, *Behav Brain Funct*, 11:23). Therefore, in order to study the general effects of PGE₂ on mating behavior in teleosts, we would be required to inject PGE₂ directly into the brain. Unfortunately, because medaka are small in size and vulnerable to head surgery, we currently have no reliable methods of administering drugs directly into the brain without affecting their subsequent performance of mating behavior (we have tried several times, but all have failed). We would be very grateful if this reviewer could understand the practical difficulties of such an experiment, although we do agree it would be of general interest.

Although it may be of interest to measure serum prostaglandin levels, the small size of medaka makes it impractical to collect sufficient amounts of serum for this measurement. Moreover, given that a previous study has shown no diurnal variation in PGE₂ levels in the medaka ovary (Fujimori *et al.*, 2011, *Mol Cell Endocrinol*, 332:67–77), we do not expect that measuring serum prostaglandin levels would yield any new insights. Instead, the expression of *ptger4b* is greatly upregulated in the ovary prior to ovulation, meaning regulation occurs at the level of the receptor (Fujimori *et al.*, 2011, *Mol Cell Endocrinol*, 332:67–77).

We agree that it may be useful to examine behaviors other than mating to determine if the behavioral effects of PGE₂ are limited to mating. This is beyond the scope of the present study and remains a topic for future research.

Reviewer #3's comment 2:

This is excellent science. However, the question remains what is its real relevance beyond the medaka? Frankly, it is not clear to me. I do agree that this work shows that certain neural (genes) machinery has been conserved across evolution but I am confused to what extent and what exactly it means. There is much greater variation in vertebrates responsiveness to prostaglandins (PGs) than the authors acknowledge (see Guillette *et al.* 2019). This study does not in my opinion reveal “fundamental truths” about how female sexual receptivity is regulated all vertebrates, but rather describes some possibilities some of which might be examples of parallel evolution.

Response to reviewer #3's comment 2:

We fully agree that it remains to be seen whether the findings obtained in medaka are applicable to other species. Given that the effects of PGE₂ on female receptivity vary greatly among species, our findings in medaka may represent only one mechanism by which PGE₂ exerts its behavioral effects. We therefore state the following in the Discussion section (line 353): “Although neurons that correspond to FeSP neurons have not been identified in other species, both NPB and EP4 have been shown to be expressed in the rat PVN^{32,33}. It would therefore be worthwhile to investigate whether the findings in medaka are applicable to other vertebrates, including rodents. The effects of PGE₂ on female receptivity vary greatly among species, however, with facilitative effects in many species, including rats and goldfish, and inhibitory effects in guinea pigs and anole lizards¹⁰. Additionally, although PGE₂/EP4 signaling has been shown to be essential for the establishment of male-typical mating behaviors in rodents¹⁹, no abnormalities were observed in the mating behavior of *ptger4b*-deficient male medaka. This suggests

that the effects of PGE₂/EP4 signaling on male sexual behavior may also differ among species. Our findings shed light on the likely mechanisms by which PGE₂ exerts different effects on the evolutionarily conserved behavioral circuitry in different species.”

To more clearly include these arguments in the manuscript, we have edited this statement as follows.

1) Line 356: “The effects of PGE₂ on female receptivity vary greatly among species, however, with facilitative effects in many species, including rats and goldfish, and inhibitory effects in guinea pigs and anole lizards¹⁰.” has been changed to “Given that the effects of PGE₂ on female receptivity vary greatly among species with facilitative effects in many species, including rats and goldfish, and inhibitory effects in guinea pigs and anole lizards¹⁰, the regulatory mechanism of female receptivity by PGE₂ found in medaka may be shared only with some species.”

2) Line 363: “Our findings shed light on the likely mechanisms by which PGE₂ exerts different effects on the evolutionarily conserved behavioral circuitry in different species.” has been changed to “Our findings provide a possible mechanism by which PGE₂ exerts different effects on the evolutionarily conserved behavioral circuitry in different species.”

Reviewer #3’s comment 3:

I will give few examples. The authors mention “rigid control of sexual receptivity by ovarian excretion of estrogen, progestins and prostaglandins.” Yet, many (thousands) of female vertebrates do not adopt sexually receptive behaviors unless males are actually present (suggesting visual, odor cues, etc. have major role - it is not just endocrine or “rigid”).

Response to reviewer #3’s comment 3:

Line 40: We recognize, of course, that female sexual receptivity depends not only on ovarian hormones, but also on multisensory cues from male conspecifics. In response, we have revised “In vertebrates, this is accomplished by the rigid control of sexual receptivity by ovarian secretion of estrogens, progestins, and prostaglandins^{1,2}.” to “In vertebrates, this is accomplished by the proper control of sexual receptivity by ovarian secretion of estrogens, progestins, and prostaglandins, in addition to appropriate sensory cues from males^{1,2}.”

Reviewer #3’s comment 4:

Further, studies of the goldfish and other cypriniform fishes show this clearly and also show that PGF2a has a key role that it is NOT associated with the ovary but rather the ovarian tract (injected egg substitutes trigger female receptivity and hyophysectomy does not influence responses – see Stacey & Liley 1974 (Nature) and Sorensen et al. 2018 (Gen. Comp Endocrinol). The cypriniformes are a group of nearly 5000 species compared to the mere 36 species found in the ricefish family to which the medaka belongs. It is hard to see it is any especially great evolutionary model (like any fish for that matter- the fishes are very diverse group!). Further, all studies of prostaglandin (PG) action in cypriniformes show that PGF2a has strong, immediate and highly specific actions on female receptivity so this seems common in this group (Stacey& Goetz 1982; Munakata & Kobayashi 2010). Notably, PGE2 seems to have no activity (Stacey & Peter 1979) and cypriniform females also are sexually inactive until PGF2a levels rise and male present and courting them (Sorensen et al. 2018). So what might the role of a strong and universal inhibitory factor such as PGE actually be and why is it needed by female fish (how might evolution favor it)?

Response to reviewer #3's comment 4:

We conclude in the present study that PGE₂/Ptger4b signaling inhibits female receptivity based on the observation that *ptger4b*-deficient females were more receptive to courting males than wild-type females. However, it may also be possible to interpret this observation as a result of reduced ability to evaluate the quality of males. The PGE₂/Ptger4b signal may delay spawning behavior to allow for the careful assessment of male suitability. However, this idea is purely speculative and not supported by empirical data, warranting verification in future studies.

Reviewer #3's comment 5:

Similar (but not identical) actions of PGFs are seen in the family cichlidae (about 2000 species) (Juntii et al. 2016) but here estradiol seems to be required as well as it is the gouramis. Variation in PG function is expected because these fishes are often nest guards and they have had well over 100 million year to evolve. The argument generally is that PGFs are produced by the ovarian tract associated with ovulation and the fact that eggs in most fishes are not viable for long after they have ovulated. PGE₂ is the precursor for PGF_{2α}, yet the possible/likely relationships between these lipids is never addressed in this manuscript either as circulatory hormones or neurotransmitters, or perhaps both?

Response to reviewer #3's comment 5:

It is true that PGF_{2α} can be synthesized from PGE₂, PGD₂, and PGH₂ by PGE 9-ketoreductase, PGD 11-ketoreductase, and 11-endoperoxide reductase, respectively. However, the activity and distribution of these enzymes in the brain have been little studied in any species, especially in teleosts. Moreover, unlike PGF_{2α}, which is derived from the ovarian tract, the brain is most likely the primary source of behaviorally relevant PGE₂. This model is consistent with the fact that brain PGE₂ levels in medaka show diurnal fluctuations with a peak at the beginning of the light period when they spawn (the present study), while ovarian PGE₂ levels do not fluctuate (Fujimori *et al.*, 2012, *Mol Cell Endocrinol*, 362:76–84). Taken together, we believe that it is best to avoid discussing the synthetic pathways of various prostaglandins in our discussion and instead focus on PGE₂.

Reviewer #3's comment 6:

Although poorly understood, a variety of responses to injected PGs have been seen in other female fishes (the vast majority of the vertebrates). The authors do state that "Of note, these prostaglandins also have facilitative effects on ... many other species, but inhibitory on others" but the authors never really account for how this could possibly be and how PGE₂ could reflect a singular conserved neuroendocrine mechanism as they suggest. Mammals diverged from fishes some 400 million years ago, show very different behaviors and physiologies, likely related to processes such as live-bearing, egg viability, internal vs external fertilization, maternal behaviors, etc.

Response to reviewer #3's comment 6:

At present, it remains uncertain what causes the diversity in the effects of PGE₂ on female receptivity. Considering our current findings and those in mammals, "The varying effect of PGE₂ on female receptivity between species may reflect differences in its site of action within behaviorally relevant circuits.", as we propose in the Discussion section (line 291). Alternatively, it may also reflect the variation of the receptors for PGE₂ and their downstream signaling pathways. In contrast to PGF_{2α} which only has a single receptor, PGE₂ has 4 metabotropic receptor subtypes in mammals, EP1, EP2, EP3, and EP4 coupled to different intracellular signalling pathways (and more in teleosts, which often have

multiple orthologs of single mammalian genes). Generally, EP1 is coupled to $G_{q/11}$ (mobilization of intracellular Ca^{2+}), EP2 and EP4 are coupled to G_s (activation of adenylyl cyclase), and EP3 is coupled to G_i (inhibition of adenylyl cyclase). If different species have different target neurons for PGE₂ and different subtypes of receptors expressed therein, the effects of PGE₂ should vary by species. However, as we state in the Discussion section (line 384), PGE₂/EP4 signaling, which was here shown to inhibit female receptivity in medaka, has been shown to be essential for the establishment of male-type mating behavior in rodents. PGE₂ can therefore exert very different behavioral effects across species, even when mediated by the same receptor subtype, suggesting that their receptor variation is not the only reason.

Line 364: In response to this and other comments from this reviewer, we have added the following text to the Discussion section: “However, the findings in medaka alone do not fully explain the divergent effects of PGE₂, and the significance of the inhibitory effects of PGE₂/Ptger4b in mature females capable of spawning remains unclear. Further comparative studies on species with different reproductive strategies are needed to address these issues.”.

Additional alterations

Additional alteration 1:

Lines 368 and 639: The section title “Materials and Methods” and subsection title “Statistical analysis” have been changed to “Methods” and “Statistics and reproducibility”, respectively, according to the journal’s guideline.

Additional alteration 2:

Line 493: “(obtained from and authenticated by Riken BRC Cell Bank)” has been added to state the source and authentication of the cell line used, according to the journal’s guideline.

Additional alteration 3:

Line 659: A “Data availability” section has been added according to the journal’s guideline. It reads: “Data availability: The data supporting the findings of this study are available within the article and its supplementary information or from the corresponding author upon reasonable request. As the raw fastq files for RNA-seq were lost due to a hard disk failure, we have provided a supplementary file describing the results of all genes in excel format. The bigwig files can be provided instead of the fastq files upon request.”.

Additional alteration 4:

All figures: All Figure panel labels have been changed to lowercase, according to the journal’s guideline.

Additional alteration 5:

The following typos have been corrected.

- 1) Lines 28 and 32: “E₂” — “E₂”
- 2) Lines 28 and 29: “ptger4b” — “ptger4b”
- 3) Line 333: “behavioural” — “behavioral”
- 4) Line 362: “sexual behavior” — “mating behavior”
- 5) Line 427: “(oryLat2 assembly) (37)” — “(oryLat2 assembly)³⁸”
- 6) Line 484: “according to⁴⁵” — “according to Vandesompele et al.⁴⁶”

- 7) Line 636: “620–700 nm” — “562–700 nm”
- 8) Line 656: “behavioural” — “behavioral”

Reviewers' comments:

Reviewer #1 (Remarks to the Author):

Additional experiments and textual revisions by the authors have greatly improved the unclear aspects of this paper. On the other hand, there are still some inadequacies that need improvements, as shown below.

1) Spontaneous firing activity of FeSP neurons

In order to prove changes in spontaneous firing in FeSP, the authors should exclude the synaptic inputs to the FeSP neuron to exclude the possibility that other neurons are responsible for the observed phenomenon. This point has not yet been remedied. TTX inhibits presynaptic transmission in low-dose bus applications, but does little to inhibit the voltage-dependent Na⁺ channel whole-cell current¹. This property of TTX can be used to prove changes in "spontaneous" firing of FeSP neurons. If TTX is difficult to apply, Tetanus Toxin, which cleaves SNARE proteins, can be used to exclude the effects from pre-synaptic inputs on FeSP neurons.

1. Wakita M, Kotani N, Akaike N. Tetrodotoxin abruptly blocks excitatory neurotransmission in mammalian CNS. *Toxicon*. 2015 Sep;103:12-8. doi: 10.1016/j.toxicon.2015.05.003. Epub 2015 May 8. PMID: 25959619.

2) Supplementary Fig.7b

The new Fig. S7b data, replaced by the authors, is consistent with the data shown in Fig. 6d for the firing pattern before stimulation with glutamate. However, after glutamate stimulation, -/- appears to have a higher firing rate than +/+. This representative data is inconsistent with the results in Fig. S7c.

3) The new supplementary Fig.7d

This result provides a remarkable information for the characterization of FeSP neurons. The involvement of ptger4b in this exceptional physiological property of FeSP neurons (a property that distinguishes them from other peptidergic neurons) is also very important in contributing to an explanation of the cause of behavioral changes in ptger4b^{-/-} mutant females. Therefore, Fig. S7d should show the results of -/- females in parallel with those of +/+ females.

Reviewer #2 (Remarks to the Author):

The authors have satisfactorily answered my comments.

Reviewer #3 (Remarks to the Author):

I have carefully read the revised manuscript. It is improved and I commend the authors for their efforts. The authors provide new detail on the experiments which are all technically sound and very interesting. Technically the entire body of work is sound. I have little doubt that prostaglandin E2 receptors play an important role mediating brain function in female medaka. Unfortunately, I continue to remain uncertain about significance of Prostaglandin E2 as a neurohormone mediating sexual receptivity in the Class Vertebrata as a whole, and rodents in particular (the implied significance of this study). The studies are of a very high quality and fascinating (i.e. strongly warrant publication), I am just unclear as to its actual significance- in my opinion the conclusions still go beyond the data in hand at present. I say this for two reasons that could and were not fully addressed in the revised manuscript:

1. The authors still lack direct evidence that prostaglandin E2 (PGE2) has the actions they suggest based on the patterns with which they find its likely receptor to be expressed. They report an inability to measure PGE2 in the brain. I appreciate that PGE2 is likely to be at extremely low (difficult to measure) levels but we are still left unfortunately with uncertainty. This is notable because of the substantial body of evidence that it is prostaglandin F2a (PGF2a) that mediates female receptivity in many other fishes from a variety of families (see below). Further, the authors do not report new data showing that they could artificially administer PGE2 to medaka to modify their behavior. Instead, they report that adding to the water was out effect but this is very unconvincing because water-borne PGs function as sex pheromones in many fishes and in any case might not be expected to diffuse through the gills (see work by Sorensen and Stacey). Intraperitoneal or brain injection of PGs are the only reasonable test, and although challenging, it has in fact, been accomplished in at least half a dozen fishes, some of which are also small (see work by Stacey). Notably, microgram quantities of PGF2a rapidly evokes female sexual receptivity in a wide variety of fishes and in also a highly specific and clearly relevant manner (see below). Thus, there is unfortunately no direct evidence of exactly what PGE2 does in the medaka although there is a lot of compelling evidence about its receptors and related evidence that they are important to brain function.

2. The authors still cannot reconcile the proposed role for PGE2 with that seen in a wide variety of other vertebrates in a convincing matter. (They add some caveats but I do not find that convincing). There appear to be contradictions and they are not explained, at least in a way that recognizes taxonomic relationships among vertebrates. In particular, there is much well-established and compelling evidence that prostaglandin F2a (PGF2a), and not PGE2, mediates sexual receptivity in a variety of fishes, and not just the goldfish (although this model is best developed). Most of this evidence comes in the form of intraperitoneal injections of PGs but in the case of goldfish this is supported by inter-cranial injections as well. This data is old but seems very solid and has been cited thousands of times. Most importantly, in the goldfish, intercranial injections of various PGs, clearly demonstrate that PGF2a, and not any PGE, most strongly and immediately induces sexual receptivity (Stacey and Peter 1979). This observation (greater activity for PGF2a than PGE2) is also supported by intraperitoneal injections (Stacey 1999, and most recently by measurements of circulating PGs in receptive females (Sorensen et al. 2018). Very notably, the goldfish has been an important model for the largest family of freshwater fishes, the Cypriniformes, which has several thousand species (unlike the medaka from the family Adrianichthyidae which has only a few dozen and some specialization for internal fertilization). A variety of Cypriniformes have also been shown to respond to PGF2a injection (Kobayashi & Stacey 1993). Perhaps even more compelling is the fact that the rapid effects of PGF2a (and not PGE2) have also been demonstrated in two other important families of fish: the Osphronemidae and the Cichlidae, the latter of which have over 1600 species and are quite distinct (evolutionarily) from the other two families. The actions of PGF2a are broad and common. (BTW, PGF metabolites and not PGEs have been implicated as sex pheromones in many fishes too (Stacey 2108), again suggesting very common usage and likely evolutionarily conserved pathways. PGF2a brain receptors have been convincingly described in this Cichlidae and PGF2a has been shown to function precisely and almost immediately, while *ptgfr* genes are found expressed in the brain, while fish show systems have been mutated for this system do not spawn (Juntii et al. 2016). The authors suggest PGF2a might be converted to PGE2 but that is not strongly supported and it is in any hard to make sense of that idea given the specificity with which PGF2a appears to act in these species. Admittedly, PGE2 has been implicated in female sexual in some (but not all) Amphibia and Mammalia (as they describe), but PGF2a has been shown to most active in at least a few Reptilia (snakes) (see older work by Crews and colleagues). It is a shame that this topic has not been reviewed well since Guillelte et al. 1991 who nevertheless describe over a dozen vertebrate species that employ PGF2a (PGE2 is only mentioned for a few species) to mediate parturient behavior in non-mammalian vertebrates. In sum, the topic of how PGs in general drive female behavior in the vertebrates is clearly very complex, and while I do not doubt at all that PGE2 has a role in the medaka, I fail to understand how I might reconcile that role with wide range of other evidence in a variety of other non-mammalian vertebrates that PGF2a has this function. I am willing to accept that understand of PG function is limited and

perhaps misconstrued but feel a really good explanation is required. The medaka is just one species amongst 36,000 fishes and nearly that number for other vertebrates (Amphibia, Reptilia, Aves and Mammalia). A far better model to understand PG function in vertebrates and how it might also function in some of their most specialized members such as the whole mammals might be ancient fishes such as lungfish. At present, I remain unconvinced that the seeming similarities between PG function in the medaka and some rodents might be an example of specialized parallel evolution, and not a common ancestral theme.

In sum, while the present study presents some extremely interesting and novel evidence that PGEs play roles in female receptivity in one species of fish, it remains to be shown exactly what that role is and its broader significance to other fishes, let alone vertebrates and rodent mammals in particular, and evolution. I hope some direct evidence might yet be produced and some explanation for how it fits in with the large body of evidence for PGF2a may yet be produced. I very much look forward to seeing this excellent work published (in whatever form it takes) although perhaps in a better informed and balanced context.

Responses to the Reviewers

First of all, we would like to thank the three anonymous reviewers for their time and effort in reviewing the revised version of our manuscript. We would also like to thank Prof. Ross Bathgate, an Editorial Board Member of *Communications Biology*, for handling our manuscript and for the opportunity to further improve the manuscript. We have considered the comments and suggestions from the reviewers #1 and #3 carefully and made necessary revisions accordingly. In the revised manuscript, revised portions are indicated in **red letters**. In this document, the original text is indicated in **blue letters** and the revised text is in **red letters**. All line numbers in this document refer to those in the revised manuscript.

Reviewer #1

Reviewer #1's overall comment:

Additional experiments and textual revisions by the authors have greatly improved the unclear aspects of this paper. On the other hand, there are still some inadequacies that need improvements, as shown below.

Response to reviewer #1's overall comment:

We thank this reviewer for their time and feedback to improve the manuscript. We have addressed each of their comments below.

Reviewer #1's specific comment 1:

1) Spontaneous firing activity of FeSP neurons

In order to prove changes in spontaneous firing in FeSP, the authors should exclude the synaptic inputs to the FeSP neuron to exclude the possibility that other neurons are responsible for the observed phenomenon. This point has not yet been remedied. TTX inhibits presynaptic transmission in low-dose bus applications, but does little to inhibit the voltage-dependent Na⁺ channel whole-cell current¹. This property of TTX can be used to prove changes in “spontaneous” firing of FeSP neurons. If TTX is difficult to apply, Tetanus Toxin, which cleaves SNARE proteins, can be used to exclude the effects from pre-synaptic inputs on FeSP neurons.

1. Wakita M, Kotani N, Akaike N. Tetrodotoxin abruptly blocks excitatory neurotransmission in mammalian CNS. *Toxicon*. 2015 Sep;103:12-8. doi: 10.1016/j.toxicon.2015.05.003. Epub 2015 May 8. PMID: 25959619.

Response to reviewer #1's specific comment 1:

We appreciate this suggestion, but we still do not believe that TTX is an appropriate tool to control for the possibility that synaptic inputs affect the observed firing of FeSP neurons, because it would more than likely also interfere with the firing of FeSP neurons themselves. The paper referred to by the reviewer (Wakita *et al.*, 2015), while of some interest, is focused only on rat hippocampal neurons, and we are not convinced that the data obtained would be reproducible in all neurons of the rat central nervous system, let alone medaka. Moreover, to our knowledge, the results of this paper have not yet been successfully adapted to a protocol for silencing presynaptic inputs while measuring spontaneous firing activity, as suggested by the reviewer. If we were to conduct the experiment suggested by the reviewer, we would have to perform a whole host of additional control experiments, including

characterization of the effects of TTX on FeSP neurons, and essentially re-perform all experiments in Wakita *et al.* (2015) on medaka. Otherwise, if a reduction in firing rate was observed after the bath application of TTX to FeSP neurons, we could not rule out the possibility that TTX simply interfered with TTX-sensitive sodium currents in FeSP neurons, rather than successfully inhibiting presynaptic inputs. Although a protocol for the selective inhibition of presynaptic inputs with TTX may one day become commonplace, at the moment, we do not believe this represents a reasonable or viable strategy to address the concerns of this reviewer.

While tetanus toxin (TeNT) is commonly used to target mammalian motor neurons to inhibit neurotransmitter release, its efficacy in other neurons and species has been less well validated. It has been reported that preincubation with TeNT for 45 minutes reduced glutamate release by only 30% in synaptosomes from the guinea pig cerebral cortex (McMahon *et al.*, 1992, *J Biol Chem*, 267:21338–21343). Thus, even if TeNT is as effective in medaka as it is in guinea pigs, it would not be sufficient as a control for synaptic inputs. Furthermore, no experimental methodology has been established for using TeNT for electrophysiological analysis in teleosts, and no single example is available in the literature that has measured spontaneous firing activity in conjunction with TeNT application in any species. Therefore, it seems unlikely that the experiment suggested by the reviewer would yield the desired results.

The standard strategy to measure spontaneous neural activity without the influence of extraneous factors is to use dissociated neurons (Bean, 2007, *Nat Rev Neurosci*, 8:451–465). This could be done by isolating and purifying FeSP neurons from the PMm/PMg, as was done for RNA-seq. Alternatively, we may measure the firing of FeSP neurons while applying various neurotransmitter antagonists (for example, AP-5/CQNX for glutamate and picrotoxin/strychnine for GABA/glycine) simultaneously or individually. These approaches are both required to provide a complete understanding of the factors regulating the firing of FeSP neurons, which will be our future challenge. Unfortunately, Dr. Mikoto Nakajo, who was responsible for the electrophysiology experiments in this paper, has left our laboratory to work in another city and does not currently have access to the equipment needed for further experiments. Any additional electrophysiological work would require him to take time away from his job and relocate back to Tokyo for an extended period. This was what we did in the previous round of revisions, but it is difficult to do this again because the above-mentioned experiments are technically demanding and time-consuming (we estimate they would take close to a year).

However, we do agree with this reviewer that we cannot rule out the possibility that synaptic inputs from other neurons may influence the firing of FeSP neurons, and we therefore propose to remove the term “spontaneous” from the entire manuscript to account for this. Please note that this change does not affect the conclusions of this study, nor does it diminish the significance of our findings. The corresponding changes are listed as follows:

- 1) Line 30: “FeSP neurons have reduced spontaneous firing activity” has been changed to “FeSP neurons have reduced firing activity”.
- 2) Line 223: “*ptger4b* deficiency reduces spontaneous firing activity” has been changed to “*ptger4b* deficiency reduces firing activity”.
- 3) Line 225: “a regular pattern of spontaneous firing” has been changed to “a regular pattern of firing”.
- 4) Line 230: “analyzed the spontaneous firing rate of FeSP neurons” has been changed to “analyzed the firing rate of FeSP neurons”.

- 5) Line 233: “a role for *ptger4b* in regulating spontaneous neuronal firing” has been changed to “a role for *ptger4b* in regulating neuronal firing”.
- 6) Line 241: “causes reduced spontaneous firing activity” has been changed to “causes reduced firing activity”.
- 7) Line 252: “no change in spontaneous firing rate” has been changed to “no change in firing rate”.
- 8) Line 308: “reduced the spontaneous firing activity of FeSP neurons” has been changed to “reduced the firing activity of FeSP neurons”.
- 9) Line 310: “the spontaneous firing of FeSP neurons” has been changed to “the firing of FeSP neurons”.
- 10) Line 319: “how the spontaneous firing activity of FeSP neurons” has been changed to “how the firing activity of FeSP neurons”.
- 11) Line 322: “increased spontaneous firing by PGE₂/Ptger4b signaling” has been changed to “increased firing by PGE₂/Ptger4b signaling”.
- 12) Line 324: “the spontaneous firing activity of FeSP neurons” has been changed to “the firing activity of FeSP neurons”.
- 13) Line 615: “Spontaneous action currents were recorded” has been changed to “Action currents were recorded”.
- 14) Line 626: “Spontaneous firing frequency was calculated” has been changed to “Firing frequency was calculated”.
- 15) Line 911: “*ptger4b* deficiency reduces spontaneous firing activity” has been changed to “*ptger4b* deficiency reduces firing activity”.
- 16) Line 986: “*ptger4b* deficiency reduces spontaneous firing activity” has been changed to “*ptger4b* deficiency reduces firing activity”.

Reviewer #1's specific comment 2:

2) Supplementary Fig.7b

The new Fig. S7b data, replaced by the authors, is consistent with the data shown in Fig. 6d for the firing pattern before stimulation with glutamate. However, after glutamate stimulation, *-/-* appears to have a higher firing rate than *+/+*. This representative data is inconsistent with the results in Fig. S7c.

Response to reviewer #1's specific comment 2:

In response to this comment, we have again replaced the trace of the *ptger4b*^{-/-} recording to better reflect the data in both Figures 6d and 7c. Please see the edited figure below.

Reviewer #1's specific comment 3:

3) The new supplementary Fig.7d

This result provides a remarkable information for the characterization of FeSP neurons. The involvement of *ptger4b* in this exceptional physiological property of FeSP neurons (a property that

distinguishes them from other peptidergic neurons) is also very important in contributing to an explanation of the cause of behavioral changes in *ptger4b*^{-/-} mutant females. Therefore, Fig. S7d should show the results of ^{-/-} females in parallel with those of ^{+/+} females.

Response to reviewer #1's specific comment 3:

The new results shown in Figure 7d demonstrate that the firing rate of FeSP neurons was not affected by acute application of either PGE₂ or an EP4 antagonist. Because the firing rate did not change in response to PGE₂ in the presence of Ptger4b, it is very unlikely to change in the absence of Ptger4b. Therefore, we do not believe that the experiment using *ptger4b* knockout females, as suggested by the reviewer, would add any significant insight. We hope that the reviewer agrees with this explanation.

Reviewer #3

Reviewer #3's overall comment:

I have carefully read the revised manuscript. It is improved and I commend the authors for their efforts. The authors provide new detail on the experiments which are all technically sound and very interesting. Technically the entire body of work is sound. I have little doubt that prostaglandin E2 receptors play an important role mediating brain function in female medaka. Unfortunately, I continue to remain uncertain about significance of Prostaglandin E2 as a neurohormone mediating sexual receptivity in the Class Vertebrata as a whole, and rodents in particular (the implied significance of this study). The studies are of a very high quality and fascinating (i.e. strongly warrant publication), I am just unclear as to its actual significance- in my opinion the conclusions still go beyond the data in hand at present. I say this for two reasons that could and were not fully addressed in the revised manuscript:

Response to reviewer #3's overall comment:

We thank this reviewer for their positive evaluation of our work. We hope we have adequately addressed their concerns below.

Reviewer #3's specific comment 1:

1. The authors still lack direct evidence that prostaglandin E2 (PGE2) has the actions they suggest based on the patterns with which they find its likely receptor to be expressed. They report an inability to measure PGE2 in the brain. I appreciate that PGE2 is likely to be at extremely low (difficult to measure) levels but we are still left unfortunately with uncertainty. This is notable because of the substantial body of evidence that it is prostaglandin F2a (PGF2a) that mediates female receptive in many other fishes from a variety of families (see below). Further, the authors do not report new data showing that they could artificially administer PGE2 to medaka to modify their behavior. Instead, they report that adding to the water was out effect but this is very unconvincing because water-borne PGs function as sex pheromones in many fishes and in any case might not be expected to diffuse through the gills (see work by Sorensen and Stacey). Intraperitoneal or brain injection of PGs are be the only reasonable test, and although challenging, it has in fact, been accomplished in at least half a dozen fishes, some of which are also small (see work by Stacey). Notably, microgram quantities of PGF2a rapidly evokes female sexual receptivity in a wide variety of fishes and in also a highly specific and clearly relevant manner (see below). Thus, there is unfortunately no direct evidence of exactly what PGE2 does in the medaka although there is a lot of compelling evidence about its receptors and related evidence that they are important to brain function.

Response to reviewer #3's specific comment 1:

We believe that there may be some confusion on the part of this reviewer. We did measure the levels of PGE₂ and its metabolites in male and female brains, and these data are presented in Fig. 3e–g in both the original and revised manuscripts. These data demonstrated that the levels of PGE₂ and its metabolites show diurnal fluctuations, being highest at the beginning of the light period when females ovulate and spawn. In contrast, PGE₂ levels in the ovary do not show diurnal changes in medaka (Fujimori *et al.*, 2011, *Mol Cell Endocrinol*, 332:67–77), leading us to conclude that the brain is most likely the primary source of behaviorally relevant PGE₂. Considering this and data from goldfish, in which a high concentration of PGE₂ injected into the third ventricle stimulated spawning behavior, but intraperitoneal injection produced only a marginal effect (Stacey 1976, *Prostaglandins*, 12:113–126; Stacey and Peter, 1979, *Physiol Behav*, 22:1191–1196), it is very likely that the import of systemic PGE₂ into the brain is limited and that intracranial injection is essential to confirm the behavioral effects of PGE₂ by *in vivo* administration experiments. We share this reviewer's concern regarding the administration of PGE₂ to water, and in fact, another study in zebrafish showed that this administration protocol had no effect on female mating behavior (Pradhan and Olsson, 2015, *Behav Brain Funct*, 11:23). Thus, neither intraperitoneal injection nor administration to water is an adequate method of administering PGE₂, and intracranial injection would be the most appropriate approach. However, as stated in the previous round of revisions, we are unable to perform this technique in medaka due to their small size and susceptibility to head injury. Much effort has been dedicated to establishing this technique in our laboratory, but thus far, we have had little success. We would like to add that, as far as we are aware, no other group is performing this technique in medaka. We apologize for this and hope this reviewer understands the situation.

Even if we were able to implement this approach, however, we believe that it would add little value to the present study. While it is true that intracranial injections of PGF_{2α} have been widely used to demonstrate its role in regulating female receptivity in teleosts, this is valid because PGF_{2α} has only a single receptor (Ptgfr); similar experiments for PGE₂ are much less valid because PGE₂ has multiple subtypes of receptors (EP1–4 in mammals and more in teleosts, which often have multiple orthologs of single mammalian genes). Even if some effect on female receptivity were observed with intracranial PGE₂ injection, it would be impossible to determine which receptor is responsible for the observed effect. It is also impossible to determine whether the effect is mediated through a single receptor or multiple receptors simultaneously, or which brain nucleus is involved in the effect. A better approach to understanding the role of PGE₂ in regulating mating behavior is via loss of function of its individual receptors, as we have done in this study. Indeed, this approach allowed us to separate the effects of PGE₂ mediated by Ptger4b from those mediated by other receptor signaling pathways. Therefore, we do not believe that the intracranial injection of PGE₂ would yield any significant results that add to the insights gained from the current series of behavioral analyses in *ptger4b*-deficient medaka, which have already documented a role for PGE₂ mediated by Ptger4b.

Reviewer #3's specific comment 2:

2. The authors still cannot reconcile the proposed role for PGE₂ with that seen in a wide variety of other vertebrates in a convincing matter. (They add some caveats but I do not find that convincing). There appear to be contradictions and they are not explained, at least in a way that recognizes taxonomic relationships among vertebrates. In particular, there is much well-established and compelling evidence that prostaglandin F_{2a} (PGF_{2a}), and not PGE₂, mediates sexual receptivity in a variety of fishes, and

not just the goldfish (although this model is best developed). Most of this evidence comes in the form of intraperitoneal injections of PGs but in the case of goldfish this is supported by inter-cranial injections as well. This data is old but seems very solid and has been cited thousands of times. Most importantly, in the goldfish, intercranial injections of various PGs, clearly demonstrate that PGF2a, and not any PGE, most strongly and immediately induces sexual receptivity (Stacey and Peter 1979). This observation (greater activity for PGF2a than PGE2) is also supported by intraperitoneal injections (Stacey 1999, and most recently by measurements of circulating PGs in receptive females (Sorensen et al. 2018). Very notably, the goldfish has been an important model for the largest family of freshwater fishes, the Cypriniformes, which has several thousand species (unlike the medaka from the family Adrianichthyidae which has only a few dozen and some specialization for internal fertilization). A variety of Cypriniformes have also been shown to respond to PGF2a injection (Kobayashi & Stacey 1993). Perhaps even more compelling is the fact that the rapid effects of PGF2a (and not PGE2) have also been demonstrated in two other important families of fish: the Osphronemidae and the Cichlidae, the latter of which have over 1600 species and are quite distinct (evolutionarily) from the other two families. The actions of PGF2a are broad and common. (BTW, PGF metabolites and not PGEs have been implicated as sex pheromones in many fishes too (Stacey 2108), again suggesting very common usage and likely evolutionarily conserved pathways. PGF2a brain receptors have been convincingly described in this Cichlidae and PGF2a has been shown to function precisely and almost immediately, while *ptgfr* genes are found expressed in the brain, while fish show systems have been mutated for this system do not spawn (Juntii et al. 2016). The authors suggest PGF2a might be converted to PGE2 but that is not strongly supported and it is in any hard to make sense of that idea given the specificity with which PGF2a appears to act in these species. Admittedly, PGE2 has been implicated in female sexual in some (but not all) Amphibia and Mammalia (as they describe), but PGF2a has been shown to most active in at least a few Reptilia (snakes) (see older work by Crews and colleagues). It is a shame that this topic has not been reviewed well since Guillette et al. 1991 who nevertheless describe over a dozen vertebrate species that employ PGF2a (PGE2 is only mentioned for a few species) to mediate parturient behavior in non-mammalian vertebrates. In sum, the topic of how PGs in general drive female behavior in the vertebrates is clearly very complex, and while I do not doubt at all that PGE2 has a role in the medaka, I fail to understand how I might reconcile that role with wide range of other evidence in a variety of other non-mammalian vertebrates that PGF2a has this function. I am willing to accept that understand of PG function is limited and perhaps misconstrued but feel a really good explanation is required. The medaka is just one species amongst 36,000 fishes and nearly that number for other vertebrates (Amphibia, Reptilia, Aves and Mammalia. A far better model to understand PG function in vertebrates and how it might also function in some of their most specialized members such as the whole mammals might be ancient fishes such as lungfish. At present, I remain unconvinced that the seeming similarities between PG function in the medaka and some rodents might is not an example of specialized parallel evolution, and not a common ancestral theme.

In sum, while the present study presents some extremely interesting and novel evidence that PGEs play roles in female receptivity in one species of fish, it remains to be shown exactly what that role is and its broader significance to other fishes, let alone vertebrates and rodent mammals in particular, and evolution. I hope some direct evidence might yet be produced and some explanation for how it fits in with the large body of evidence for PGF2a may yet be produced. I very much look forward to seeing this excellent work published (in whatever form it takes) although perhaps in a better informed and balanced context.

Response to reviewer #3's specific comment 2:

We believe that the reviewer has misinterpreted the conclusions of our study. We do not claim that PGE₂ is the primary prostaglandin regulating female receptivity in medaka or that the functions of PGE₂ and PGF_{2α} are mutually exclusive. We strongly agree with this reviewer that PGF_{2α} plays an important role in the regulation of female receptivity in teleosts. In fact, recent unpublished data from our laboratory show that female medaka deficient for *ptgfr* are completely unreceptive to male courtship. This result clearly indicates that PGF_{2α} is essential for female receptivity in medaka, as has been reported in other teleost species, but because this is an ongoing area of research in our laboratory, and that this paper focuses on Ptger4b rather than Ptgfr, we do not wish to include this result in the manuscript. Our overall results for PGE₂ and PGF_{2α} reveal that both of these prostaglandins play a role in regulating female receptivity. That is, while PGF_{2α} probably acts as a hormonal trigger for spawning behavior after ovulation, PGE₂ most likely acts as a neuromodulator synthesized locally in the brain and inhibits female receptivity when binding to Ptger4b. We hope that this clarification has adequately addressed the concerns of this reviewer.

We would like to add that the aim of the current study was to elucidate the regulatory mechanism, role in mating behavior, and mode of action of *ptger4b*, which was identified as being expressed in an ovarian-dependent manner in medaka FeSP neurons. We certainly agree that there is a lack of data on possible crosstalk between the PGE₂ and PGF_{2α} pathways and interspecies conservation of PGE₂ effects on mating behavior, but these topics are beyond the scope of the current study and will be addressed in the forthcoming papers.

We would also like to correct a misstatement made by the reviewer. We did not make the claim anywhere in our manuscript or responses that PGF_{2α} is converted to PGE₂. In the previous round of reviews, this reviewer highlighted that PGE₂ can serve as a precursor to PGF_{2α}, to which we agreed that such crosstalk may exist in the medaka brain, but we also responded that this possibility has been little studied and should be avoided to prevent overcomplicating the discussion. Please find below the reviewer's original comment and our response:

Reviewer #3's comment 5:

Similar (but not identical) actions of PGFs are seen in the family cichlidae (about 2000 species) (Juntii et al. 2016) but here estradiol seems to be required as well as it is the gouramis. Variation in PG function is expected because these fishes are often nest guards and they have had well over 100 million year to evolve. The argument generally is that PGFs are produced by the ovarian tract associated with ovulation and the fact that eggs in most fishes are not viable for long after they have ovulated. PGE₂ is the precursor for PGF_{2α}, yet the possible/likely relationships between these lipids is never addressed in this manuscript either as circulatory hormones or neurotransmitters, or perhaps both?

Response to reviewer #3's comment 5:

It is true that PGF_{2α} can be synthesized from PGE₂, PGD₂, and PGH₂ by PGE 9-ketoreductase, PGD 11-ketoreductase, and 11-endoperoxide reductase, respectively. However, the activity and distribution of these enzymes in the brain have been little studied in any species, especially in teleosts. Moreover, unlike PGF_{2α}, which is derived from the ovarian tract, the brain is most likely the primary source of behaviorally relevant PGE₂. This model is consistent with the fact that brain PGE₂ levels in medaka show diurnal fluctuations with a peak at the beginning of the

light period when they spawn (the present study), while ovarian PGE₂ levels do not fluctuate (Fujimori *et al.*, 2012, *Mol Cell Endocrinol*, 362:76–84). Taken together, we believe that it is best to avoid discussing the synthetic pathways of various prostaglandins in our discussion and instead focus on PGE₂.

Regarding the suggestion about lungfish, it is true that they may be a better model than medaka for understanding the evolutionary origin of prostaglandin functions in mammals. However, since this is not the purpose of our study, we are unable to include lungfish as an additional model in this paper. We will leave this idea for future research.

Finally, we would like to address this reviewer's concern about our arguments based on comparisons with rodents. Our conclusions have considered data across species and highlighted both differences and similarities in the regulation of mating behavior by PGE₂ in vertebrates. This research area is still in its infancy; for example, there are no studies detailing the expression sites of prostaglandin receptors or neuropeptide B in the brains of other teleost species. We therefore had to rely on data from rodents, from which the majority of available information on PGE₂ has been derived, for many of our conclusions. We agreed with this reviewer's earlier concerns and have toned down our interpretation of the results accordingly. In the revised manuscript, we conclude that the effects of PGE₂ on mating behavior vary across species and that further studies are warranted to understand whether the effects of PGE₂ on female receptivity and its mode of action found in this study are conserved. We do not believe that these conclusions are unreasonable or overstate the significance of our data.

Additional alterations

Additional alteration 1:

We have found that the y-axis titles for the graphs in Supplemental Figs. 4 and 5, which show the male behavioral data, were mislabeled with titles for the female behavioral data. We have edited these y-axis titles as follows.

Panel c: “number of wrappings refused” has been changed to “number of wrapping failures”.

Panel d: “% individuals that were approached by male” has been changed to “% individuals that approached female”.

Panels e a and f: “% individuals that received courtship display” has been changed to “% individuals that performed courtship display”.

We apologize for this oversight. Please see the edited figures below.

Supplementary Fig. 4

Supplementary Fig. 5

Additional alteration 2:

Line 377: The following text has been added to the Methods section in response to the comment from the editorial office that we should include a statement that shows ethical approval.

“All animal procedures were performed in accordance with the guidelines of the Institutional Animal Care and Use Committee of the University of Tokyo. The committee requests the submission of an animal-use protocol only for use of mammals, birds, and reptiles, in accordance with the Fundamental Guidelines for Proper Conduct of Animal Experiment and Related Activities in Academic Research Institutions under the jurisdiction of the Ministry of Education, Culture, Sports, Science and Technology of Japan (Ministry of Education, Culture, Sports, Science and Technology, Notice No. 71; June 1, 2006). Accordingly, we did not submit an animal-use protocol for this study, which used only teleost fish and thus did not require approval by the committee.”

Reviewers' comments:

Reviewer #1 (Remarks to the Author):

Regarding this revised manuscript, the description of the experimental results and the figures have been sufficiently improved. The answers to my requests have been satisfactorily accomplished. I commend the authors for their patient efforts.

Reviewer #3 (Remarks to the Author):

This manuscript has improved with its revision and the authors are once again to be commended for their diligence. However, once again the manuscript unfortunately falls short of being clear and transparent. I was pleased to see how they addressed reviewer #1's comments but extremely disappointed that my comments about the confusion I had over the roles of PGF2a and PGE2 in the medaka were not addressed at all in the revised manuscript. The work is very interesting and it is a shame and unnecessary that its main finding (that PGE2 functions as neuromodulator of peptidergic neurons in the medaka preoptic area that controls female sexual receptivity) is difficult to understand for anyone without considerable expertise in the details of how neuromodulators function in the vertebrate brain. As the authors' state in their comments to the editor- I had indeed misunderstood much of the manuscript but I cannot believe I will be the only one unless it is changed. The following statements by the authors are especially noteworthy. "We believe that the reviewer has misinterpreted the conclusion of the study, we do not claim that PGE2 is the primary prostaglandin regulating female receptivity in medaka or that the functions of PGE2 or PGF2a are mutually exclusive. We strongly agree with this reviewer that PGF2a plays an important role ... the primary aim of the current study was to elucidate the regulatory mechanisms, role in mating behavior, and mode of action of ptger4b. ... That is while PGF2a probably acts as a hormonal trigger for spawning behavior after ovulation, PGE2 most likely acts as a neuromodulator synthesized locally in the brain and inhibits receptivity when binding to Ptger4b." I am very pleased to see these statements and do agree with them all but strongly believe that the manuscript would be greatly improved if they were included in the revised manuscript to prevent others from also reaching misunderstandings.

Please consider the following changes which I truly believe will greatly improve the value of this manuscript.

1. Please consider changing the title to "A prostaglandin E2 receptor Ptger4b modulates the activity of female-specific neurons that control female sexual receptivity in a model fish." (The present title strongly suggests that that Ptger4b regulates female sexual receptivity in all vertebrates which I now know is not true and had confused me)
2. The following terms are presently all used to describe the effects of PGs on female sexual behavior through the manuscript: "regulate", "mediate", "facilitate", "modulate." I found this very confusing. Please review how these terms are used, pick one (modulate seems best maybe), define it, and use it alone.
3. Please embellish and correct your introduction to PGs. In the Introduction, line 51-53, the authors state "In several teleost species, gonadal estrogens are not essential for female receptivity, and instead prostaglandins E2a and PGF2a facilitate female receptivity (1.9)." This is simply not correct and very confusing because it is vague. First, none of these studies actually suggest that PGE2 has a role mediating female behavior (the brain injection work found PGE2 to be 10 times less effective than PGF2a which was interpreted as lack of specificity); indeed, all studies discuss how PGF2a alone functions as an endocrine mediator of female sexual receptivity while Sorensen et al. 2018 (not referenced) proves it by measuring PGF2a in the blood of ovulated, sexually receptive goldfish. Second, it is unclear what is meant by "facilitate." I strongly suggest rewording this statement along

the following lines: " In half a dozen teleost species, gonadal estrogens are known not be essential to female receptivity while blood-borne PGF2a mediates female sexual receptivity; however, whether and how this receptivity might be modulated centrally including by other prostaglandins, is not known." This statement could bring great clarity I think.

4. Please revise the discussion to mention that while it is an accepted fact that PGF2a functions as primary endocrine driver of sexual receptivity in many teleost fish, and presumably medaka, this activity now appear to modulated by peptidergic neurons modulated by PGE2. Please add a short paragraph to state some of the interesting unknowns (ex. whether there might be cross-talk between PGE2 and PGF2a, the role PGF2s in medaka, the role of inhibition, whether the processes noted in medaka and seen in other fishes including some ancient ones, etc.). This will attract interest and increase clarity.

In conclusion, the authors have an excellent study, it would be shame if they ignored one reviewer who asks for more explanation and clarity that will only make it more readily comprehensible to a wider audience. I would like to see this manuscript published soon.

Responses to the Reviewers

First of all, we would like to thank the anonymous reviewers for their time and effort in reviewing the revised version of our manuscript. We would also like to thank Prof. Ross Bathgate, an Editorial Board Member of *Communications Biology*, for handling our manuscript and for the opportunity to further improve the manuscript. We have considered the comments and suggestions from reviewer #3 and made necessary revisions accordingly. In the revised manuscript, revised portions are indicated in **red letters**. In this document, the original text is indicated in **blue letters** and the revised text is in **red letters**. All line numbers in this document refer to those in the revised manuscript.

Reviewer #3

Reviewer #3's overall comment:

This manuscript has improved with its revision and the authors are once again to be commended for their diligence. However, once again the manuscript unfortunately falls short of being clear and transparent. I was pleased to see how they addressed reviewer #1's comments but extremely disappointed that my comments about the confusion I had over the roles of PGF2a and PGE2 in the medaka were not addressed at all in the revised manuscript. The work is very interesting and it is a shame and unnecessary that its main finding (that PGE2 functions as neuromodulator of peptidergic neurons in the medaka preoptic area that controls female sexual receptivity) is difficult to understand for anyone without considerable expertise in the details of how neuromodulators function in the vertebrate brain. As the authors' state in their comments to the editor—I had indeed misunderstood much of the manuscript but I cannot believe I will be the only one unless it is changed. The following statements by the authors are especially noteworthy. “We believe that the reviewer has misinterpreted the conclusion of the study, we do not claim that PGE2 is the primary prostaglandin regulating female receptivity in medaka or that the functions of PGE2 or PGF2a are mutually exclusive. We strongly agree with this reviewer that PGF2a plays an important role ... the primary aim of the current study was to elucidate the regulatory mechanisms, role in mating behavior, and mode of action of pterg4b. ... That is while PGF2a probably acts as a hormonal trigger for spawning behavior after ovulation, PGE2 most likely acts as a neuromodulator synthesized locally in the brain and inhibits receptivity when binding to Pterg4b.” I am very pleased to see these statements and do agree with them all but strongly believe that the manuscript would be greatly improved if they were included in the revised manuscript to prevent others from also reaching misunderstandings.

Please consider the following changes which I truly believe will greatly improve the value of this manuscript.

In conclusion, the authors have an excellent study, it would be a shame if they ignored one reviewer who asks for more explanation and clarity that will only make it more readily comprehensible to a wider audience. I would like to see this manuscript published soon.

Response to reviewer #3's overall comment:

We apologize that you found the manuscript confusing. We have carefully followed your comments and made revisions accordingly that, we hope, will provide sufficient explanation and clarity so that the manuscript is readily comprehensible to a wider audience.

Reviewer #3's specific comment 1:

Please consider changing the title to “A prostaglandin E2 receptor Ptger4b modulates the activity of female-specific neurons that control female sexual receptivity in a model fish.” (The present title strongly suggests that that PTger4b regulates female sexual receptivity in all vertebrates which I now know is not true and had confused me)

Response to reviewer #3's specific comment 1:

In response to this reviewer's concern, the manuscript title has been changed to indicate the model organism used in our study (medaka).

Line 1: “Prostaglandin E₂ receptor Ptger4b regulates female-specific peptidergic neurons and female sexual receptivity” has been changed to “Prostaglandin E₂ receptor Ptger4b regulates female-specific peptidergic neurons and female sexual receptivity in medaka”.

Reviewer #3's specific comment 2:

The following terms are presently all used to describe the effects of PGs on female sexual behavior through the manuscript: “regulate”, “mediate”, facilitate”, “modulate.” I found this very confusing. Please review how these terms are used, pick one (modulate seems best maybe), define it, and use it alone.

Response to reviewer #3's specific comment 2:

The use of synonyms in scientific English, where appropriate, is commonplace to avoid writing dull, repetitive, lifeless text. This is not an abstract concept that would be unfamiliar to the average reader of *Communications Biology*. Replacing these verbs with a single verb (modulate) would do little improve the clarity of the manuscript, outside of a select few people without knowledge of their definitions. We would also argue that the suggestion put forth by this reviewer would instead decrease the overall clarity and readability of the manuscript, as each of these verbs differ slightly in their meaning and are important for contextual nuance, as described below.

“Regulate” is commonly used in biology as a term meaning to exert control over a certain process. The words regulate/regulation/regulatory are used a total of 6 times throughout the manuscript to describe the effects of prostaglandins on mating behavior, as follows.

Line 1: Prostaglandin E₂ receptor Ptger4b regulates medaka female-specific peptidergic neurons and female sexual receptivity.

Line 46: However, little information is available on the neural basis of female sexual receptivity in non-rodent species, and moreover, accumulating evidence highlights large variations in the hormonal regulation of vertebrate mating behavior across taxa.

Line 110: Considering that PGE₂ has been implicated in the regulation of female sexual receptivity across vertebrate phyla¹⁰, but its neural mechanisms of action remain poorly understood, we selected this gene as the focus of our analysis.

Line 272: Collectively, FeSP neurons may integrate ovarian signaling and multiple different neuropeptide signaling pathways to regulate female receptivity.

Line 286: Taken together, PGE₂/Ptger4b signaling may act downstream of E₂/Esr2b signaling in FeSP neurons to regulate female receptivity.

Line 372: Given that the effects of PGE₂ on female receptivity vary greatly among species with facilitative effects in many species, including rats and goldfish, and inhibitory effects in guinea pigs and anole lizards¹⁰, the regulatory mechanism of female receptivity by PGE₂ found in medaka may be

shared only with some species.

“Mediate” means to bring about a result, often indirectly through another thing. This term is used throughout the manuscript, mostly as a way to say that something (such as Ptger4b) is a downstream effector of something else (such as estrogen). For example, line 313: *Considering that ptger4b expression in FeSP neurons requires ovarian estrogens, it is probable that the estrogen-dependent electrophysiological activity of these neurons is **mediated**, at least in part, by PGE₂/Ptger4b signaling.* The term “mediate” is not used to describe the effects of prostaglandins on female mating behavior in the manuscript.

“Facilitate” means to make (an action or process) easy or easier. It is similar to regulate; however, it is only appropriate to use in cases of positive effects. The words facilitate/facilitative are used 3 times to describe the effects of prostaglandins on mating behavior, as follows.

Line 51: *In several other teleost species, including goldfish (*Carassius auratus*) and African cichlid (*Astatotilapia burtoni*), gonadal estrogens are not essential for female sexual receptivity, and instead, prostaglandin F_{2α} (PGF_{2α}) **facilitates** female receptivity^{1,9}.*

Lines 53: *Although it is not known whether other prostaglandin species, such as prostaglandin E₂ (PGE₂), also centrally regulate female receptivity in teleosts, PGE₂ has been shown to have **facilitative** effects on female receptivity in rats and hamsters but have inhibitory effects in guinea pigs and anole lizards (*Anolis carolinensis*)¹⁰.*

Lines 372: *Given that the effects of PGE₂ on female receptivity vary greatly among species with **facilitative** effects in many species, including rats and hamsters, and inhibitory effects in guinea pigs and anole lizards¹⁰, the regulatory mechanism of female receptivity by PGE₂ found in medaka may be shared only with some species.*

“Modulate” means to exert a changing/controlling influence over something. The term “modulation” is only used once in the entire manuscript (see below), but not to describe the effects of prostaglandins on female mating behavior.

Line 75: *This effect was likely due to **modulation** of the electrophysiological properties of these neurons by Ptger4b, which ultimately govern neuropeptide release.*

One can plainly see that only the terms “regulate” and “facilitate” are used to describe the effects of prostaglandins on female mating behavior, in contrast to reviewer #3’s claim. We consistently used the term “regulate” to describe our own results, and as a generic term meaning “to exert control over” because both positive and negative effects of prostaglandins on female mating behavior were reported in the literature. “Facilitate”, on the other hand, was used exclusively to describe a positive effect of prostaglandins on female receptivity, according to its dictionary definition. We do not believe these are recondite terms that would require specialized knowledge to differentiate, and the average reader should have no trouble understanding their usage in the manuscript, provided they have sufficient knowledge of English.

We would like to add that the majority of authors on this publication have Japanese as a first language, and none of them reported any difficulty interpreting and understanding the usage of these verbs as written, leading us to believe that the writing in the manuscript is appropriate, even for readers who are non-native English speakers.

Reviewer #3's specific comment 3:

Please embellish and correct your introduction to PGs. In the Introduction, line 51-53, the authors state “In several teleost species, gonadal estrogens are not essential for female receptivity, and instead prostaglandins E₂ and PGF₂α facilitate female receptivity (1.9).” This is simply not correct and very confusing because it is vague. First, none of these studies actually suggest that PGE₂ has a role mediating female behavior (the brain injection work found PGE₂ to be 10 times less effective than PGF₂α which was interpreted as lack of specificity); indeed, all studies discuss how PGF₂α alone functions as an endocrine mediator of female sexual receptivity while Sorensen et al. 2018 (no referenced) proves it by measuring PGF₂α in the blood of ovulated, sexually receptive goldfish. Second, it is unclear what is meant by “facilitate.” I strongly suggest rewording this statement along the following lines: “In half a dozen teleost species, gonadal estrogens are known not to be essential to female receptivity while blood-borne PGF₂α mediates female sexual receptivity; however, whether and how this receptivity might be modulated centrally including by other prostaglandins, is not known.” This statement could bring great clarity I think.

Response to reviewer #3's specific comment 3:

We agree with this reviewer that the evidence provided for the facilitative effect of PGE₂ on female receptivity in goldfish is rather weak. In response to this comment, the text in the Introduction section has been edited as follows to emphasize the lack of knowledge on the effects of PGE₂ on female receptivity in teleosts and instead highlight its effects in other species.

Line 51: “In several other teleost species, including goldfish (*Carassius auratus*) and African cichlid (*Astatotilapia burtoni*), gonadal estrogens are not essential for female sexual receptivity, and instead, prostaglandins E₂ (PGE₂) and F₂α facilitate female receptivity^{1,9}. Of note, these prostaglandins also have facilitative effects on female receptivity in many other species but have inhibitory effects in guinea pigs and anole lizards (*Anolis carolinensis*)¹⁰. These lines of evidence suggest that there may be a degree of underlying variation—at either the structural or chemical level—in behaviorally relevant circuits across species.” has been changed to “In several other teleost species, including goldfish (*Carassius auratus*) and African cichlid (*Astatotilapia burtoni*), gonadal estrogens are not essential for female sexual receptivity, and instead, prostaglandin F₂α (PGF₂α) facilitates female receptivity^{1,9}. Although it is not known whether other prostaglandin species, such as prostaglandin E₂ (PGE₂), also centrally regulate female receptivity in teleosts, PGE₂ has been shown to have facilitative effects on female receptivity in rats and hamsters but have inhibitory effects in guinea pigs and anole lizards (*Anolis carolinensis*)¹⁰. These lines of evidence suggest that there may be a degree of underlying variation—at either the structural or chemical level—in behaviorally relevant circuits and the action of hormonal mediators therein across species.”

The concluding paragraph of the Discussion section has also been edited to reflect the changes to the introduction.

Line 372: “Given that the effects of PGE₂ on female receptivity vary greatly among species with facilitative effects in many species, including rats and goldfish, and inhibitory effects in guinea pigs and anole lizards¹⁰, the regulatory mechanism of female receptivity by PGE₂ found in medaka may be shared only with some species.” has been changed to “Given that the effects of PGE₂ on female receptivity vary greatly among species with facilitative effects in many species, including rats and hamsters, and inhibitory effects in guinea pigs and anole lizards¹⁰, the regulatory mechanism of female receptivity by PGE₂ found in medaka may be shared only with some species.”

Reviewer #3's specific comment 4:

Please revise the discussion to mention that while it is an accepted fact that PGF_{2a} functions as primary endocrine driver of sexual receptivity in many teleost fish, and presumably medaka, this activity now appear to modulated by peptidergic neurons modulated by PGE₂. Please add a short paragraph to state some of the interesting unknowns (ex. whether there might be cross-talk between PGE₂ and PGF_{2a}, the role PGF_{2s} in medaka, the role of inhibition, whether the processes noted in medaka and seen in other fishes including some ancient ones, etc.). This will attract interest and increase clarity.

Response to reviewer #3's specific comment 4:

In response to this comment, the following paragraph has been added to the Discussion section:

Line 347: Our study adds to the prevailing view that prostaglandins are major regulators of female receptivity in teleosts, demonstrating a role for PGE₂ in addition to PGF_{2α}. However, these two prostaglandin species differ greatly in their mode of action. PGF_{2α} is essential for female receptivity and is derived from the ovary, where the presence of recently ovulated eggs in the ovarian tract triggers the production and release of PGF_{2α}, which then acts as a hormonal signal of reproductive status to the brain to induce mating behaviors^{1, 9, 33, 34}. In contrast, PGE₂ levels in the ovary do not show diurnal changes corresponding to reproductive state¹⁶, meaning that the diurnal fluctuations of PGE₂ and its metabolites in the brain—which are highest at the beginning of the light period when females spawn—are not derived from the ovary. Considering this and data from zebrafish (*Danio rerio*), which showed PGE₂ had no effect on mating behavior when administered to the water³⁵, it is probable that the import of systemic PGE₂ into the brain is limited, and that the brain is the primary source of behaviorally relevant PGE₂. Thus, PGE₂ most likely acts as a locally synthesized neuromodulator in the brain and inhibits female receptivity when binding to Ptger4b. It is not currently clear what the functional significance of the inhibitory nature of PGE₂/Ptger4b signaling is to female mating behavior. It may be that PGE₂/Ptger4b delays spawning to allow for the careful assessment of male suitability/mate choice, but further work is required to validate this idea.

The following references, which are cited in the above text, have been added to the reference list (33–35):

33. Sorensen, P.W., Appelt, C., Stacey, N.E., Goetz, F.W. & Brash, A.R. High levels of circulating prostaglandin F_{2α} associated with ovulation stimulate female sexual receptivity and spawning behavior in the goldfish (*Carassius auratus*). *Gen. Comp. Endocrinol.* **267**, 128–136 (2018).

34. Stacey, N.E. & Liley, N.R. Regulation of spawning behavior in the female goldfish. *Nature* **247**, 71–72 (1974).

35. Pradhan, A. & Olsson, P.E. Zebrafish sexual behavior: role of sex steroid hormones and prostaglandins. *Behav. Brain Funct.* **11**, 23 (2015).

As a consequence, references 33–51 have been renumbered to 36–54 in the text and in the reference list.

We hope these revisions provide the additional clarity sought by this reviewer.

Additional alterations

We have identified and corrected some notational errors in the manuscript, as follows:

1) Line 134: “Both the mutations of the ERE-like sequences at positions -1076 and -714 abolished the

E₂ induction of luciferase activity” has been changed to “Both mutations of the ERE-like sequences at positions -1076 and -714 abolished E₂'s induction of luciferase activity”.

2) Line 387: “dr-R” has been changed to “d-rR”.

3) Line 742: “Fujimori, C., Ogiwara, K., Hagiwara, A., & Takahashi, T. New evidence for the involvement of prostaglandin receptor EP4b in ovulation of the medaka, *Oryzias latipes*. *Mol. Cell. Endocrinol.* **332**, 76–84 (2012).” has been changed to “Fujimori, C., Ogiwara, K., Hagiwara, A. & Takahashi, T. New evidence for the involvement of prostaglandin receptor EP4b in ovulation of the medaka, *Oryzias latipes*. *Mol. Cell. Endocrinol.* **332**, 76–84 (2012).”

4) Line 897: “Diurnal fluctuations of *ptger4b* expression in the male and female brains” has been changed to “Diurnal fluctuations of *ptger4b* expression in male and female brains.”

5) Line 902: “(d, g)” has been change to “(d-g)”.

We apologize for these oversights.

REVIEWERS' COMMENTS:

Reviewer #3 (Remarks to the Author):

I thank the authors for their thoughtful responses and for editing the manuscript to increase its clarity. I find the changes to be significant improvements and am very pleased, especially the change to the title and the addition of new information on PGF2a. I have only two relatively minor comments that I would like them to consider but leave that to them.

1) Although I do not disagree with how the authors presently use the terms "regulate", "facilitate" and "modulate", these words actually are not synonyms (Oxford English Dictionary) and are also commonly misused; it would improve manuscript clarity were they to be defined (perhaps in parentheses) in the manuscript. This may be an example of how non-English speakers use the English language more correctly than native English speakers who may never have learned the technical definitions.

2) In the revised discussion, the authors now state that "PGF2a ... is derived from the ovary." This statement is debatable: Stacey&Liley (1974) and Sorensen et al. (2018) both show that injection of non-egg material into the oviducts (not the ovary) of goldfish stimulates PGF2a production and female receptivity. (just something interesting to consider)

In sum, I thank the authors for this excellent work (and patience) and look forward to seeing it in press.